# Ferritin triggers neutrophil extracellular trap-mediated cytokine storm through Msr1 contributing to adult-onset Still's disease pathogenesis

Hyperferritinemic syndrome, an overwhelming inflammatory condition, is characterized by high ferritin levels, systemic inflammation and multi-organ dysfunction, but the pathogenic role of ferritin remains largely unknown. Here we show in an animal model that ferritin administration leads to systemic and hepatic inflammation characterized by excessive neutrophil leukocyte infiltration and neutrophil extracellular trap (NET) formation in the liver tissue. Ferritin-induced NET formation depends on the expression of peptidylarginine deiminase 4 and neutrophil elastase and on reactive oxygen species production. Mechanistically, ferritin exposure increases both overall and cell surface expression of Msr1 on neutrophil leukocytes, and also acts as ligand to Msr1 to trigger the NET formation pathway. Depletion of neutrophil leukocytes or ablation of Msr1 protect mice from tissue damage and the hyperinflammatory response, which further confirms the role of Msr1 as ferritin receptor. The relevance of the animal model is underscored by the observation that enhanced NET formation, increased Msr1 expression and signalling on neutrophil leukocytes are also characteristic to adult-onset Still's disease (AOSD), a typical hyperferritinemic syndrome. Collectively, our findings demonstrate an essential role of ferritin in NET-mediated cytokine storm, and suggest that targeting NETs or Msr1 may benefit AOSD patients.

Hyperferritinemic syndrome, an overwhelming inflammatory condition, is characterized by high serum ferritin level and sustained by excessive release of pro-inflammatory cytokines, leading to a cytokine storm[1]. It encompasses four clinical conditions, including adult-onset Still's disease (AOSD), macrophage activation syndrome (MAS), catastrophic anti-phospholipid syndrome (CAPS), and septic shock, all of which are burdened by multi-organ failure and high mortality rate[2]. Recently, severe coronavirus disease-19 (COVID-19) has been recognized as a fifth member of this spectrum due to similar clinical and laboratory features[3]. Understanding the potential mechanisms triggering the development of a pernicious inflammatory loop in cytokine storm is critical to make targeted therapeutics. Ferritin is an iron storage protein and preserved throughout species[4]. It was previously thought to be an acute-phase protein only as a biomarker of systemic inflammation, but there is accumulating evidence that hyperferritinemia is not only a consequence of the inflammatory process but also a critical part of the pathogenic mechanism[5]. Regulated by pro-inflammatory cytokines, ferritin can further promote expression of pro-inflammatory mediators, providing interesting insights on its association with cytokine storm[6]. To date, the mechanism of how ferritin plays a pathogenic role in facilitating the inflammatory burden in hyperferritinemic syndrome remains to be determined.

✉e-mail: bjj@njmu.edu.cn; jingwang@shsmu.edu.cn; yangchengde@sina.com; huqiongyi131@163.com

In fact, ferritin is largely over-expressed in AOSD patients[7]. AOSD, a systemic inflammatory disorder triggered by virus infections, is characterized by spiking fever, evanescent rash, arthralgia or arthritis and hepatosplenomegaly[8,9]. Liver involvement is common in AOSD, ranging from mild hepatitis to life-threatening hepatic failure[10]. Hallmarks of AOSD are high levels of ferritin and innate immune cell hyperactivation, with neutrophils being the most abundant[11]. The involvement of neutrophils in the pathogenesis of AOSD has been widely acknowledged and investigated[12]. Our previous study has demonstrated the enhanced ability of neutrophil extracellular trap (NET) formation in AOSD patients, which is associated with the leukocyte immunoglobulin-like receptor A3 gene, a genetic risk factor for AOSD[13]. We also reported that circulating NETs can serve as potential biomarkers for identifying liver injury and response to glucocorticoid in AOSD[14]. NETs are extracellular web-like structures that are released by neutrophils and decorated with histones, proteinase, and granular proteins[15]. As potential drivers of amplified inflammatory storm in AOSD, neutrophils crosstalk with macrophages to activate NLRP3 inflammasome by NETs, leading to a vicious loop accelerating inflammatory response[16]. However, the precise mechanisms regulating NET formation in AOSD have not been completely elucidated.

Since its discovery, dysregulated NET formation has extensively been investigated in the initiation and progression of autoimmune and autoinflammatory diseases[17–19]. The molecular pathways to release NETs depend on the diverse inflammatory context of each disease, leading us to hypothesize the potential role of ferritin as a driver of NET formation in hyperferritinemic syndrome[20]. With emerging recognition of the pathogenic role of ferritin as a pro-inflammatory mediator in cytokine storm, there is now a compelling reason to explore ferritin-NETs interplay. A recent study has also demonstrated that macrophage scavenger receptor 1 (Msr1), a pattern recognition receptor to recognize a broad spectrum of ligands, binds ferritin and may facilitate the internalization of ferritin[21]. It has been reported that Msr1 promotes fulminant hepatitis by enhancing NETs[22]. We thus hypothesized that Msr1 might be involved in the ferritin-neutrophil interaction.

Although high ferritin levels have always been considered as a sign of hyperinflammation, their pathogenic roles in triggering cytokine storm remains elusive. In the present study, we show that ferritin, as a ligand to scavenger receptor Msr1, promotes inflammatory response by inducing NET formation in a peptidylarginine deiminase 4 (PAD4), neutrophil elastase (NE), and reactive oxygen species (ROS)-dependent way. Overall, these data reveal a mechanism through which high ferritin levels contribute to neutrophil-mediated inflammation and highlight the importance of NETs and Msr1 as therapeutic targets in hyperferritinemic syndrome.

## Results

### Ferritin leads to systemic and hepatic inflammation with increased neutrophil infiltration

To investigate the pathogenetic role of ferritin in inflammation, we administrated ferritin intraperitoneally to mice. Ferritin-treated mice developed a marked hepatosplenomegaly at 6 h post injection (Fig. 1a). Administration of ferritin also resulted in neutrophilia and subsequently a mild increase in monocyte numbers in peripheral blood (Fig. 1b; Supplementary Fig. 1). Consistent with hyperferritinemic syndrome, ferritin-treated mice developed elevated levels of serum pro-inflammatory cytokines after 3 h, including interleukin (IL)−6 and tumor necrosis factor (TNF)-α. There was also an elevation of IL-10 at 3 h, monocyte chemoattractant protein (MCP)−1 at 3–12 h, and interferon (IFN)-γ at 12 h (Fig. 1c).

Abnormal liver function is very common in the spectrum of the hyperferritinemic syndrome. Ferritin-treated mice developed hepatic inflammation with abundant inflammatory cell infiltration (Fig. 1d). In addition, ferritin-treated mice developed mildly increased serum ALT

levels, markedly elevated by 6 h and persisted beyond 12 h post injection (Fig. 1e). We also observed an elevation of inflammatory genes (Fig. 1f), suggesting that ferritin contributes to liver injury by enhancing hepatic inflammation. We next explored infiltrated immune cell in the liver of ferritin-treated mice. Neutrophil infiltration in the liver from ferritin-treated mice started at 3 h, reaching the peak at 6 h post injection (Fig. 1g; Supplementary Fig. 2). Particularly, early neutrophil recruitment was accompanied by monocyte infiltration (Fig. 1g). Macrophages began to rise at 12 h and rose significantly at 24 h. Percentage of T cells and B cells was not statistically elevated (Fig. 1g). As confirmed by immunochemistry, livers from mice treated with ferritin for 6 h showed an enhanced neutrophilic infiltration (Fig. 1h; Supplementary Fig. 3). Taken together, these data indicate that ferritin drives systemic and hepatic inflammation, mainly manifesting as excessive release of cytokines, neutrophilia, hepatosplenomegaly, and neutrophil infiltration in the liver.

### Ferritin activates neutrophil to generate NETs

Given the critical role of NETs in neutrophil-mediated inflammation, we examined neutrophil NET formation in ferritin-treated mice. After ferritin injection, bone marrow-derived neutrophils (BMDN) showed augmented NET formation at 3 h and most robust at 6 h (Supplementary Fig. 4a). We further isolated BMDNs from ferritin-treated mice for 6 h, and stimulated them with LPS or PMA ex vivo. BMDNs from ferritin-treated mice for 6 h exhibited an increased capacity of releasing extracellular DNA (Fig. 2a, b; Supplementary Fig. 4b). Moreover, the liver tissue from ferritin-treated mice for 6 h manifested a significantly increased expression of NET markers, NE and citrullinated histone H3 (citH3) measured by western blotting (WB) (Fig. 2c). In line with this observation, enhanced NET formation was displayed by immunofluorescence staining of NE, citH3, and myeloperoxidase (MPO) (Fig. 2d). To track and identify NETs after ferritin injection, we used intravital imaging experiments with a combination staining approach: a specific neutrophil marker (Ly6G), extracellular DNA dye (Sytox Green) and a NET marker (NE). In ferritin-treated mice, a robust increased recruitment of neutrophils (Ly6G+) into the liver and elevated extracellular DNA and NE were observed (Fig. 2e; Supplementary Movies 1–2). Taken together, these data suggest a contribution of ferritin on NETosis on top of its effects on neutrophil recruitment into the liver.

### Neutrophils are essential for ferritin-induced inflammation

After confirming that neutrophils were increased in blood and liver of ferritin-treated mice, we investigated their roles in ferritin-mediated hyperinflammatory response. Since mice treated with ferritin for 6 h showed the most severe inflammatory phenotype and highest degree of liver damage, we selected these mice as models for subsequent experiments. We applied anti-Ly6G antibody to deplete neutrophils. The anti-Ly6G antibody efficiently depleted circulating and hepatic neutrophils in ferritin-treated mice, but hepatic monocytes were enhanced (Fig. 3a, b). Next, we assessed the effects of neutrophil depletion on the systemic inflammation and hepatic injury, and found that anti-Ly6G antibody ameliorated the degrees of hepatomegaly in ferritin-treated mice (Fig. 3c). Meanwhile, mice with anti-Ly6G antibody had lower systemic inflammation as demonstrated by diminished serum levels of IL-6, TNF-α, and MCP-1 (Fig. 3d). Serum levels of IL-10 and IFN-γ showed no significant difference (Fig. 3d). Further, mice with neutrophil depletion had reduced NE and citH3 expression and less NET structures in the livers (Fig. 3e, f). haematoxylin and eosin (H&E) staining confirmed that neutrophil depletion reduced ferritin-induced liver inflammatory infiltration (Fig. 3g). Reduction in *Il1b*, *Il6*, and *Tnfa* mRNA levels in the liver tissue homogenates and serum ALT levels were also seen in mice with neutrophil depletion (Fig. 3h, i). Together, these observations suggest that neutrophils are required for ferritin-triggered systemic inflammation and liver injury.

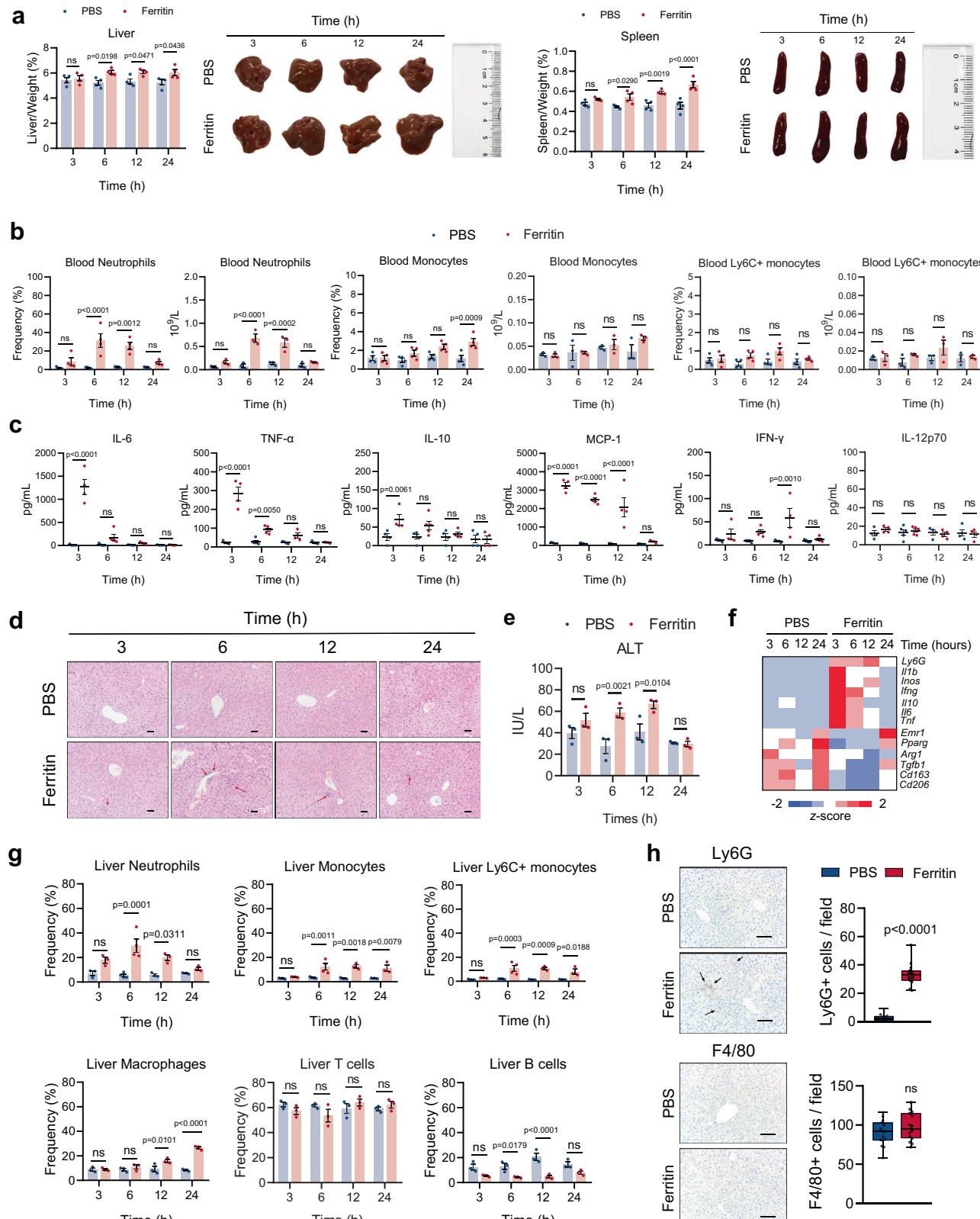

## Ferritin-induced NETosis is dependent on PAD4, NE and ROS

Next, the ability of ferritin to promote NET formation was confirmed in neutrophil from healthy donors. Ferritin with concentration from 10 nM to 1000 nM significantly facilitated NET formation in control neutrophils, with 100 nM ferritin having the best stimulation effect (Fig. 4a). The protein level of citH3 and the activity of NE were gradually increased with ferritin stimulation (Fig. 4b, c). Further,

intracellular ROS levels were increased both in ferritin-treated human neutrophils and BMDNs from ferritin-treated mice at 6 h (Fig. 4d). Next, we pre-treated neutrophils with Cl-amidine (a PAD4 inhibitor), Sivelestat (a NE inhibitor) or DPI (a NADPH oxidase inhibitor), prior to stimulation with 100 nM ferritin. All inhibitors abrogated ferritin-induced NET formation as confirmed by cell-free DNA, MPO-DNA and immunofluorescence staining of NET structures (Fig. 4e, f).

**Fig. 1 | Ferritin induces systemic inflammation and liver injury in vivo.** Mice were treated with ferritin for the indicated period of time. **a** Liver and spleen weight normalized to body weight (*n* = 4), and representative images of the livers and spleens are shown. **b** Dynamic changes of the frequencies and numbers of peripheral myeloid cells were assessed by flow cytometry (*n* = 4 for cell frequencies, *n* = 3 for cell numbers). **c** Serum levels of IL-6, TNF-α, IL-10, MCP-1, IFN-γ, and IL-12p70 were measured by CBA analysis (*n* = 5 in 6 h groups, *n* = 4 in other groups). **d** Livers from control and ferritin-treated mice were subjected to H&E staining. Scale bars, 50 μm. Red arrowheads indicate inflammatory infiltrations. One representative image of livers from three mice in each group was shown. **e** Serum ALT levels at each time point were assessed (*n* = 3). **f** Heatmap displaying the expression levels of genes involved in inflammatory response. The mean relative mRNA levels of 4 biological replicates were calculated and the data were represented as z score. **g** Flow cytometry analysis for myeloid cell infiltration in the liver (*n* = 4 in 6 h groups for neutrophils, monocytes and Ly6C+ monocytes, *n* = 3 in other groups). **h** Neutrophil (Ly6G) and macrophage (F4/80) infiltrations in the livers from ferritin-treated mice for 6 h were stained by immunohistochemistry, and the percentage of positive cells was quantified (5 fields per mouse, *n* = 3). Representative sections are shown. Scale bars, 50 μm. Red arrowheads indicate neutrophil infiltrations. Data are presented as means ± SEM except for box and whiskers with minima to maxima in graph (**h**); ns not significant. Unpaired two-sided Student's *t* test (**h**) and two-way ANOVA with Bonferroni's multiple comparison test (**a**–**c**, **e** and **g**). Source data are provided as a Source Data file.

After demonstrating the indispensable roles of PAD4, NE, and ROS in ferritin-induced NET formation, we assessed the effects of these inhibitors in mice. Cl-amidine, sivelestat, and DPI did not affect the number of circulating neutrophils, but ameliorated hepatic neutrophil infiltration and the degree of hepatomegaly in ferritin-treated mice (Fig. 4g, h; Supplementary Fig. 5a). Decreased serum levels of IL-6, TNF-α, and MCP-1 were also observed (Fig. 4i; Supplementary Fig. 5b). Importantly, Cl-amidine, sivelestat, and DPI significantly blocked NET formation in the livers determined by WB analysis and intravital imaging (Fig. 4j, k). Meanwhile, liver inflammation cells infiltration, mRNA levels of *Il1b*, *Il6*, and *Tnfa*, and serum ALT levels were significantly attenuated (Fig. 4l–n). To confirm the hypothesis that PAD4, NE and ROS-mediated NET formation contributed to ferritin-induced inflammation, we transplanted BM from wild type (WT), Padi4−/−, Elane−/− or Cybb−/− mice into WT mice and gave them ferritin injection (Fig. 5a). As expected, BMDNs from mice transplanted with Padi4−/−, Elane−/− or Cybb−/− BM displayed lower levels of NET release after ferritin injection (Fig. 5b). We also observed less NET deposition in the livers of mice transplanted with Padi4−/−, Elane−/− or Cybb−/− BM after ferritin injection (Fig. 5c, d). Deletion of these genes resulted in decreased peripheral neutrophil frequency and hepatic neutrophil infiltration (Fig. 5e; Supplementary Fig. 6a), reduced hepatomegaly (Fig. 5f), and suppression of serum cytokine levels and liver inflammation (Fig. 5g–j; Supplementary Fig. 6b). In summary, these results indicate that ferritin promotes NET formation and inflammatory response in a PAD4, NE, and ROS-dependent way.

### Ferritin promotes NET formation by activating Msr1 on neutrophils

To further investigate the molecular changes of neutrophil in ferritin-treated mice, we performed RNA sequencing (RNA-seq) of BMDNs. By gene ontology analysis, we found genes involved in inflammatory response, such as innate immune response and cytokine production, were enriched in the BMDNs from ferritin-treated mice (Fig. 6a). We then focused on the expression levels of potential ferritin receptors and found Msr1 was significantly increased (Fig. 6b). This change was confirmed in the FACS-sorted BMDNs by quantitative real-time PCR (qRT-PCR) (Fig. 6c; Supplementary Fig. 7a). WB confirmed the augmented protein level of Msr1 in BMDNs (Fig. 6e). The mRNA and protein levels of Msr1 in the livers were also significantly increased (Fig. 6d, e). Flow cytometry revealed that CD204+ neutrophils were increased both in the peripheral blood and liver in mice with ferritin treatment (Fig. 6f; Supplementary Fig. 7b, c). Similarly, ferritin stimulation led to enhanced Msr1 expression on cell surface in human neutrophils (Fig. 6g; Supplementary Fig. 7d). Fucoidan, a Msr1 antagonistic ligand, reduced the ability of human neutrophils to release NETs in response to ferritin (Fig. 6h, i). In line with this result, ferritin promoted neutrophil to generate NETs from WT mice but not Msr1−/− mice (Fig. 6j). Msr1−/− mice were then exposed to systemic ferritin, and NET formation of BDMNs and NET infiltration within the liver were measured. A significant reduction in the capacity to form NETs spontaneously was observed, and the ability to generate NET in response to LPS and PMA

was also attenuated (Fig. 6k, l). In addition, a dramatic reduction in the NET infiltration within the liver was validated by WB and confocal intravital imaging (Fig. 6m, n). Collectively, these results reveal that Msr1, as a receptor of ferritin on neutrophils, is critical for ferritin-induced NET formation.

### Msr1 ablation ameliorates ferritin-induced inflammation

We then investigated the role of Msr1 in ferritin-induced inflammation in vivo. Though Msr1 ablation did not affect neutrophil number in blood (Fig. 7a), Msr1−/− mice with ferritin treatment displayed less neutrophil infiltration in the liver and lower degrees of hepatomegaly compared with WT mice (Fig. 7b, c). Besides, serum levels of IL-6, TNF-α, and MCP-1 were significantly attenuated in Msr1−/− mice than WT mice, while serum levels of IL-10 and IFN-γ were not affected (Fig. 7d). Consistently, a dramatic reduction in the inflammatory cell infiltration within the liver of ferritin-treated Msr1−/− mice was observed by H&E staining, a result supported by qRT-PCR, where a significant diminished expression levels of *Il1b*, *Il6*, and *Tnfa* were detected (Fig. 7e, f). In line with this observation, Msr1 deficiency led to attenuated liver damage with lower ALT activity (Fig. 7g). Because of the important role of mitogen-activated protein kinases (MAPK) and Akt in the initiation of NETosis, we examined the expression of these kinases in human neutrophils upon ferritin stimulation. Significant enhanced levels of ERK, JNK, and p38 phosphorylation were observed in ferritin-treated neutrophils, but not Akt phosphorylation (Supplementary Fig. 8), suggesting activating MAPK pathways in human neutrophils in response to ferritin. Notably, BMDNs from ferritin-treated Msr1−/− mice displayed lower levels of MAPKs phosphorylation compared with ferritin-treated WT mice (Fig. 7h). Collectively, these results indicate that Msr1 on neutrophils is critical for ferritin-triggered hyperinflammatory conditions and liver injury.

### Hyperferritinemia contributes to increased NET formation in AOSD patients

AOSD, an overwhelming systemic inflammatory disorder, is characterized by high serum ferritin and neutrophil hyperactivation with enhanced NET formation, we then investigated whether ferritin could play a pathogenic role in AOSD through NET formation. A positive correlation was observed between ferritin and circulating NETs, including serum cell-free DNA, citH3-DNA, MPO-DNA, and NE-DNA levels (Fig. 8a). To further determine whether serum ferritin could mediate NET release, neutrophils from healthy controls were treated with sera from 3 patients with active AOSD with ferritin either absorbed away (after absorption) or not absorbed (before absorption). The capacity of sera to promote NET formation was significantly attenuated after absorption of ferritin (Fig. 8b). To obtain evidence of NET formation in the liver of AOSD patients, we analyzed the liver biopsy from an AOSD patient with ongoing liver injury. H&E staining revealed mild hepatitis with neutrophilic infiltration and immunofluorescence analysis demonstrated the deposition of NETs in areas of neutrophilic infiltration (Fig. 8c). Next, we explored the potential of ferritin-Msr1 pathway to promote NET formation in AOSD neutrophils. We observed

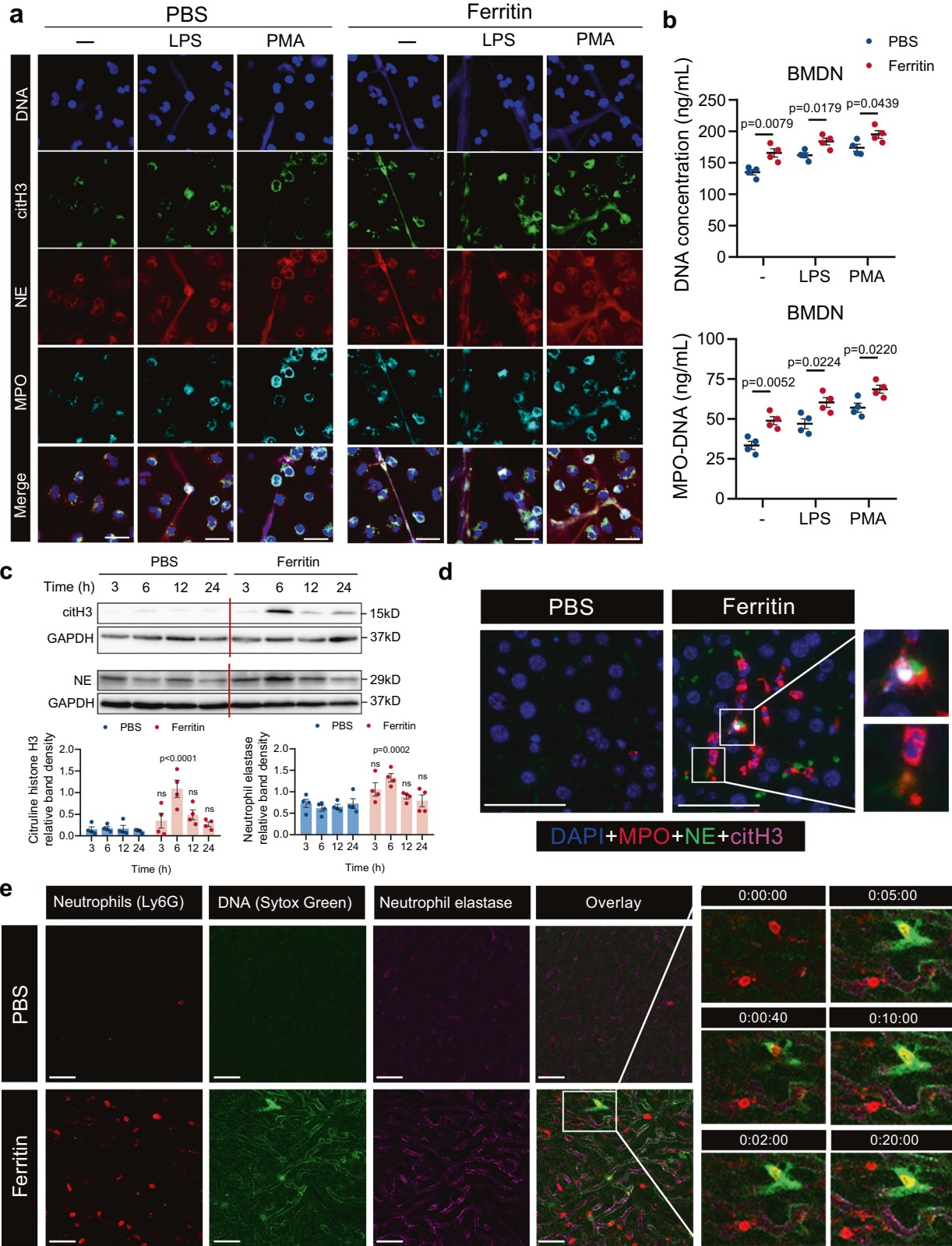

that the mRNA and protein levels of Msr1 as well as Msr1 expression on neutrophils' membrane were significantly increased (Fig. 8d–f; Supplementary Fig. 9). In AOSD patients, neutrophils showed increased basal level of NET formation compared with that in health controls (Fig. 8g). And neutrophils in response to ferritin showed a slight increase in NET formation, probably due to overactivated neutrophils in AOSD (Fig. 8h). Furthermore, using Msr1 antagonistic ligand, NET

release was significantly reduced (Fig. 8h, i), suggesting the cardinal role of Msr1 in ferritin-NETosis in patients with AOSD.

## Discussion

Due to the catastrophic outbreak of COVID-19, the fifth type of hyperferritinemic syndrome, much attention has been paid on the pathogenetic role of ferritin in cytokine storm sustained by a vicious pro-

**Fig. 2 | Ferritin promotes neutrophils to generate NETs in vivo. a** Representative immunofluorescence staining of spontaneous, LPS- and PMA-induced NET formation in the BMDNs from ferritin-treated mice at 6 h. Neutrophils stained with antibodies to citH3 (green), NE (red), and MPO (cyan), and with the DNA dye Hoechst (blue). Scale bars, 20 μm. **b** Quantification of cell-free DNA and MPO-DNA complexes of BMDNs from ferritin-treated mice at 6 h (*n* = 4). **c** Western blotting analysis for NET markers citH3 and NE in the liver from ferritin-treated mice (*n* = 4). **d** Immunofluorescence detection of NE (green), citH3 (magenta), MPO (red) and DNA (blue) in situ in the liver of ferritin-treated mice at 6 h. One representative image of livers from three mice per group was shown. Scale bars, 20 μm. **e** Representative images of neutrophil infiltration (red) and NET release (DNA: green; NE: magenta) in the liver of ferritin-treated mice at 6 h by intravital microscopy. One representative image of livers from two mice per group was shown. Scale bars, 50 μm. Data are presented as means ± SEM; unpaired two-sided Student's *t* test (**b**) and two-way ANOVA with Bonferroni's multiple comparison test (**c**). Source data are provided as a Source Data file.

inflammatory loop. Using murine models and clinical samples, we comprehensively assess the pro-inflammatory effect of ferritin. We show here that ferritin activates neutrophils to form NETs and contribute to cytokine storm and liver inflammation. Ferritin activates Msr1 of neutrophils to facilitate ERK, JNK, and p38 activation, thus enhancing neutrophil infiltration in the liver with abundant NET formation. Importantly, we also reveal that treatment with Msr1 inhibitor or genetic loss of Mrs1 result in a significant reduction in neutrophil infiltration in the liver with ameliorated systemic inflammatory response.

Cytokine storm was first used to describe the pathogenesis of graft-versus-host disease (GVHD) and subsequently demonstrated to be associated with various infectious, autoimmune, and inflammatory diseases[23]. Cytokine storm is characterized by increased production of IL-1β, IL-6, IL-10, IFN-γ, TNF-α, and other cytokines. These inflammatory mediators activate immune system and lead to a life-threatening uncontrollable inflammation[24]. However, the understanding of cytokine storm is still in the early stage. Recently, COVID-19, a virus-induced respiratory disease has brought attention to the cytokine storm[25]. In severe COVID-19 patients, a hyperinflammatory status with a massive release of pro-inflammatory cytokines is proved, which shows similarity to hyperferritinemic syndrome. Interestingly, neutrophilia and hyperferritinemia predict poor outcomes in patients with severe COVID-19[26]. Thus, some studies have considered severe COVID-19 as a fifth member of the spectrum of hyperferritinemic syndrome[3,27]. AOSD, triggered by viral infections[8], also belongs to hyperferritinemic syndrome. In our previous study, a significant higher serum level of ferritin was observed in active AOSD patients in comparison with severe COVID-19[28].

Ferritin was traditionally regarded as a conserved protein with the mere function of iron storage. However, increasing evidence supports the idea that high circulating ferritin may not only reflect an acute-phase response, but also play a critical role in inflammatory response[29]. It has been revealed that ferritin regulates NF-κB activation and subsequent expression of pro-inflammatory molecules in an iron-independent way[30]. In addition, the deletion of ferritin heavy chain ameliorates the inflammatory burden in the model of sepsis, including reduction of IL-1β, IL-6, IL-12, and IFN-γ and improves survival[31]. Moreover, ferritin can induce Toll-like receptor (TLR) 9 expression and other TLRs in a leukemic cell line[5]. Herein, we confirmed the pathogenetic role of ferritin in vivo. We found systemic inflammation in ferritin-treated mice, characterized by cytokine storm and liver damage, which is similar to clinical manifestations of the spectrum of hyperferritinemic syndrome[2].

AOSD, a rare systemic autoinflammatory disorder, is typically characterized by hyperferritinemia and cytokine storm[32]. After the first trigger, there is an amplification of inflammation activated by innate immune system, leading to uncontrollable inflammatory cascade reaction. Our previous study has revealed an enhanced NET formation to link neutrophils and macrophages[16]. It is noteworthy that NETs can facilitate the production of inflammatory mediators and further be enhanced by these mediators, leading to a vicious uncontrollable, inflammatory loop[33]. Although it has yet to be determined whether NETs contribute to the amplified inflammatory process in AOSD patients, there is accumulating evidence to indicate inflammatory cytokines such as IL-1β, IL-18, and IL-6 in the AOSD milieu, which can

interact with NETs[34]. Indeed, a NETs-cytokine loop exists in various disease, including COVID-19, atherosclerosis and systemic lupus erythematosus (SLE)[35–37]. Despite recent advances in exploring the role of neutrophil and NET in the pathophysiology of AOSD, little is known about underlying mechanism. Herein, our observation that high ferritin levels correlates with increased circulating NETs levels in patients with AOSD and ferritin promotes neutrophils to release NETs in an animal model and human primary cells implies that ferritin may play an important role in the initiation of NET formation. Although we have not tested the importance of ferritin for NET formation in COVID-19, the ferritin-NETs-cytokine storm loop may be validated in the future since enhanced NET formation and hyperferritinemic state are features of patients with severe COVID-19[28,36].

Considering that many endogenous pathways are involved in NETosis, we investigated the underlying molecular mechanisms of ferritin-induced NET formation[38]. Our previous studies have already shown that the generation of ROS is needed for NET formation in AOSD[16]. Besides, several studies have suggested that PAD4 inhibitor and NE inhibitor prevent NETosis in human neutrophils and in mice[39,40]. In contrast, it has been noted that NETosis induced by different physiological stimuli is very diverse in the engaged pathways. Indeed, granulocyte-macrophage colony-stimulating factor (GM-CSF) and TNF could induce NETosis in a ROS-independent but PAD4-dependent way[41]. In this regard, our study analyzed these biological processes in ferritin-stimulated neutrophils and revealed that ROS, NE, and PAD4 are essential molecules for ferritin-induced NET formation. Furthermore, MAPKs and Akt are two important signalling pathways that have been closely linked to NETosis. This study shows that ERK, JNK, and p38 are activated, but Akt is not affected in neutrophils after ferritin stimulation.

The receptor of ferritin on neutrophils has been marginally addressed in the past decades. Recently, Msr1 was shown to interact directly with extracellular ferritin[21]. Msr1, a membrane-bound scavenger receptor on macrophages and monocytes that recognize various targets, exerts a regulatory function in inflammatory response and cytokine production[42,43]. In a murine model of fulminant hepatitis, Msr1 acts as an inflammatory accelerator through activating neutrophils and promoting the release of NETs, supporting the important pathogenetic role of Msr1 in neutrophilic inflammation[22]. However, the physiological effect of their interaction remains unknown. In our current work, we reveal that Msr1 is expressed on neutrophils from AOSD patients and ferritin-treated mice, and accompanied by p38, ERK, and JNK pathway activation. Using pharmacological and genetic approaches, Msr1 is found to be indispensable for ferritin-induced NET release and subsequent inflammatory response. The mechanism of how ferritin is internalized by Msr1 will be investigated in our further study.

Liver involvement is very common in AOSD. The most frequent manifestation of AOSD is mild hepatitis, but liver failure with hepatic necrosis can be life-threatening in several cases[9]. A major histologic finding of the liver from AOSD is the infiltration of neutrophils. Our previous study has shown that neutrophil-derived lipocalin-2 could serve as a potential biomarker to identify liver injury of AOSD and evaluate the severity of liver dysfunction in AOSD patients[44]. Excessive activation of neutrophils and NETs are believed to promote liver injury in many other liver diseases[45]. In virus-induced fulminant hepatitis,

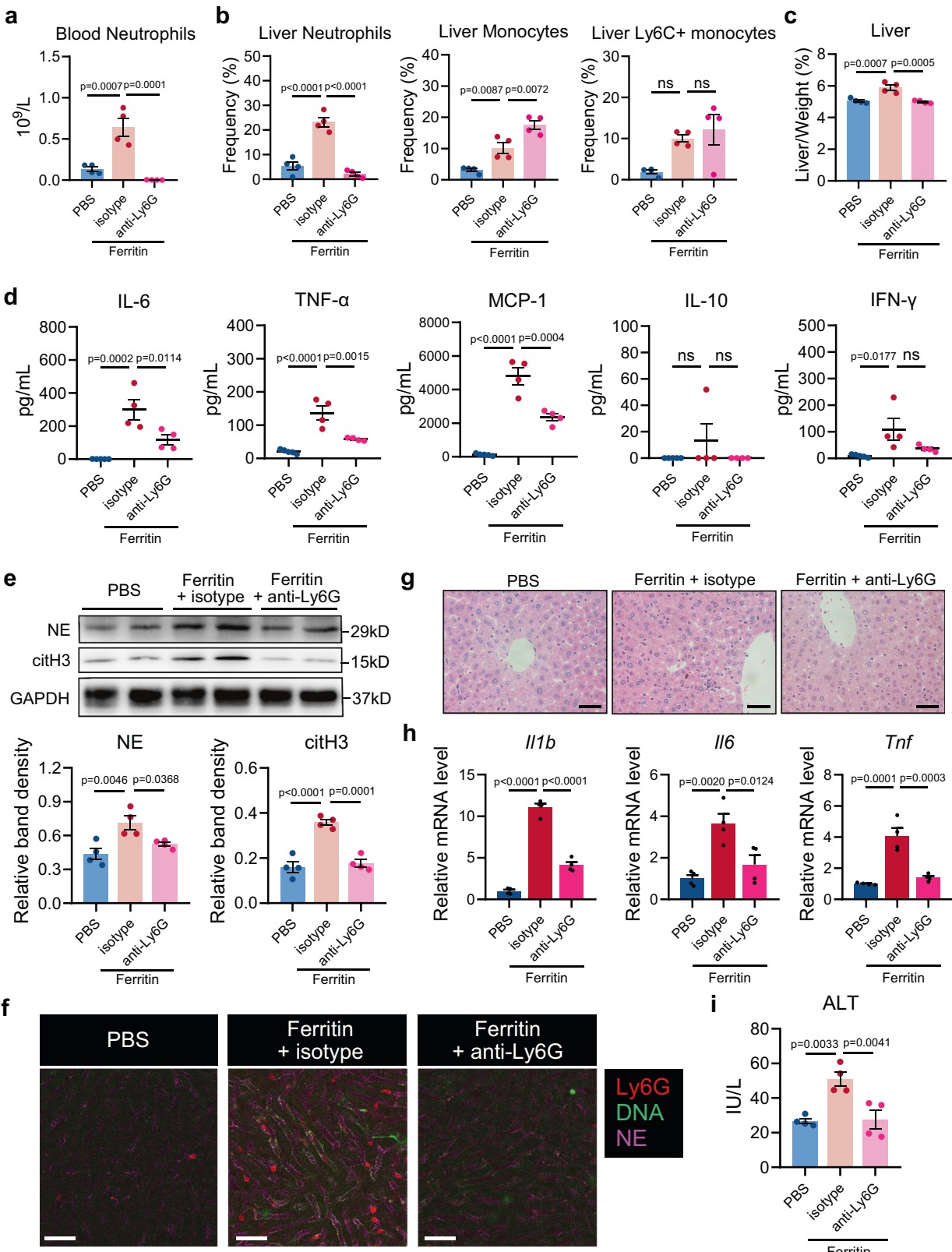

enhanced NETosis resulted in complement activation and subsequent cytokine production[22]. In addition, it has been described that NETs exacerbate inflammatory cascades and sterile inflammation in liver ischemia and reperfusion injury[46]. Decreased levels of NET clearance also contributed to liver injury in alcoholic liver disease[47]. In our present study, we observed that ferritin treatment induced hepatic neutrophilic inflammation, which was ameliorated by neutrophil depletion

and suppression of NET formation. This indicates a dominant role of NETs as inflammatory mediators in hyperferritinemia-related hepatic inflammation.

There are several limitations in our study. First, the ferritin used in human and mouse experiments is from equine spleen, which is composed of 24 subunits in variable ratios of heavy and light chain encoded by the ferritin heavy chain (FTH) and ferritin light chain (FTL)

**Fig. 3 | Neutrophils are required for ferritin-induced inflammation.** Neutrophils were depleted by anti-Ly6G antibody in ferritin-treated mice at 6 h. **a, b** Peripheral neutrophil numbers (**a**), and liver myeloid cell infiltration (**b**) were measured by flow cytometry (*n* = 4). **c** Changes of liver to body weight ratio were shown (*n* = 4). **d** Serum cytokines IL-6, TNF-α, IL-10, MCP-1 and IFN-γ were evaluated by CBA analysis (*n* = 5 in PBS group, *n* = 4 in isotype and anti-Ly6G groups). **e** Western blotting analysis for NET markers citH3 and NE in the liver (*n* = 4). **f** Representative images of neutrophils infiltration (red) and NET release (DNA: green; NE: magenta) in the liver by intravital microscopy. One representative image of livers from two mice per group was shown. Scale bars, 50 μm. **g** Liver inflammatory infiltration was detected by H&E staining. One representative image of livers from four mice in each group was shown. Scale bars, 50 μm. **h, i** Liver mRNA expression of *Il1b*, *Il6* and *Tnfa* was measured by qRT-PCR (**h**) and serum ALT level was detected (**i**) (*n* = 4). Data are presented as means ± SEM; ns not significant. one-way ANOVA with Bonferroni's multiple comparison test. Source data are provided as a Source Data file.

genes, respectively. We did not test other sources of ferritin, such as human/mouse ferritin or heavy/light chain ferritin, for functional validation. Second, we demonstrated the pathogenic role of ferritin in an animal model and to some extent in patients of AOSD. Additional research is needed to verify the link between ferritin and neutrophils in other hyperferritinemic conditions.

In conclusion, our findings demonstrate the underlying relationship between hyperferritinemia and NET formation. Ferritin induces the release of NETs in a Msr1-dependent way, which then contributes to systemic and hepatic inflammation. Accordingly, abolishing the NETs or Msr1 could abrogate the ferritin-induced hyperinflammatory process. Our study highlights the important role of ferritin-Msr1-NETs pathway in the overwhelming inflammatory response, serving as a therapeutic target against the spectrum of hyperferritinemic syndrome.

## Methods
Human biological samples were obtained under a protocol approved by the Institutional Research Ethics Committee of Ruijin Hospital (ID: 2016–62), Shanghai, China. All biological samples from patients and healthy donors were collected after obtaining informed consent from all participants. All experimental animal protocols described in this study were approved by the Animal Care Committee of Shanghai Jiao Tong University School of Medicine.

### Human participants
A total of 64 AOSD patients (45 active and 19 in active AOSD patients) admitted to the Department of Rheumatology and Immunology, Ruijin Hospital from May 2017 to December 2018 were consecutively included in the present study, and serum samples were collected from all participants. All patients fulfilled Yamaguchi's criteria after exclusion of those with infectious, neoplastic and autoimmune disorders. All serum samples were stored at −80 °C immediately after collection. The AOSD disease activity of each patient was assessed using a modified Pouchot score[48]. The biological samples of AOSD patients and healthy donors were obtained under a protocol approved by the Institutional Research Ethics Committee of Ruijin Hospital (ID: 2016–62), Shanghai, China., and all the participants provided informed consent. Supplementary Table 1 shows the main characteristics of AOSD patients at the time of the blood sampling.

### Mice
Female, 8–12 weeks WT FVB/n (#215), and C57BL/6 (#219) mice were purchased from Vital River Laboratories (Beijing, China). Msr1-deficient (Msr1$^{-/-}$) mice on a C57BL/6 background were provided from Prof. Jingjing Ben in Nanjing Medical University (Nanjing, China), which were purchased from Jackson Laboratory (#006096, RRID: IMSR_JAX: 006096)[49]. Padi4$^{-/-}$, Elane$^{-/-}$ and Cybb$^{-/-}$ mice were obtained from Shanghai Model organisms (Padi4: #NM-KO-190334, Elane: #NM-KO-201544, Cybb: #NM-KO-18031). Animals were maintained under pathogen-free conditions and housed with no more than five animals per cage under a 12 h light/dark cycle with free access to mouse chow and water, ambient temperature 22–24 °C and humidity 50–70%. All experiments were performed on sex-and 8 to 12-week-old age-matched animals. Ferritin (F4503, sterile-filtered, from equine spleen, composed of heavy chains and light chains,

Sigma-Aldrich, USA) was injected intraperitoneally (i.p.) to WT FVB/n mice at a dose of 60 μg/g of body weight. After treatment for 3, 6, 12 and 24 h, mice were anesthetized with isoflurane inhalation for body weight measurement and blood collection, and then euthanized by 2.5% chloral hydrate (0.1 mL/10 g) and rapid cervical dislocation for liver and spleen collection. The size of liver and spleen were measured. The blood and serum were harvested for flow cytometry and cytokine analysis. Liver samples were collected, fixed in 4% paraformaldehyde and embedded in paraffin. Additional liver samples were stored at −80 °C.

In some experiments, to deplete neutrophil, rat anti-Ly6G antibody (clone 1A8, BE0075, BioXcell, 200 μg/mouse, i.p.) was given 24 h and 2 h prior to ferritin injection. Rat IgG2a isotype control (BE0089, BioXcell) was administered in the same way. To inhibit NET formation, Cl-amidine (20 mg/kg, HY-100574A, MCE, China, i.p.) was given 24 h and 1 h before ferritin injection, DPI (1 mg/kg, D2926, Sigma-Aldrich, USA, i.p.) was given 0.5 h before ferritin injection and sivelestat (50 mg/kg, HY-17443, MCE, China, i.p.) was given 1 h after ferritin injection. For exploring the effect of Msr1 in ferritin-NETs pathway, female, 8–12 weeks Msr1-deficient (Msr1$^{-/-}$) C57BL/6 mice were injected with ferritin (60 μg/g) intraperitoneally. Female, 8–12 weeks WT C57BL/6 mice were used as controls. All experimental protocols described in this study were approved by the Animal Care Committee of Shanghai Jiao Tong University School of Medicine.

### Bone marrow transplant
For performing bone marrow transplants (BMTs), the recipient mice (C57BL/6 wild-type mice, female, 8 weeks old) were lethally irradiated with 10 Gy from a caesium gamma source. Donor bone marrow cells (5 × 10$^6$ cells) obtained from WT, Padi4$^{-/-}$, Elane$^{-/-}$ and Cybb$^{-/-}$ donor mice (female, 8–12 weeks old) were intravenous injected into the recipient mice (after irradiation). Four weeks after BMT, the recipient mice were injected with ferritin (60 μg/g) intraperitoneally for further experiments.

### Isolation and stimulation of human neutrophils
Briefly, heparinized blood from AOSD patients and healthy controls was isolated by density gradient centrifugation on Polymorphprep (AS1114683, Axis-Shield, Dundee, UK) for 40 min at 400 × g without braking. The median layer containing neutrophils/red blood cells (RBCs) was transferred to a fresh tube. The neutrophil/RBC pellet was suspended in RBC lysis buffer (Servicebio) and neutrophils were washed in sterile PBS and suspended. Neutrophils (1 × 10$^6$ cells/mL) were cultured in RPMI 1640 (Hyclone) supplemented with 10% fetal bovine serum (FBS) and were stimulated with PMA (20 nM, P1585, Sigma-Aldrich) or ferritin (10–1000 nM, F4503, Sigma-Aldrich) for 3.5 h at 37 °C.

In some assays, to inhibit NET formation, neutrophils were pre-treated with PAD4 inhibitor Cl-amidine (10 μM, HY-100574A, MCE, China), DPI (25 μM, D2926, Sigma-Aldrich, USA), sivelestat (20 μM, HY-17443, MCE, China) or fucoidan (100 μg/mL, 20357, Cayman chemical company, MI). In some experiments, to determine whether ferritin in plasma could mediate NETosis or not, we absorbed ferritin protein away from sera samples by incubating with ELISA plates from ferritin ELISA kit (CSB-E05187h, CUSABIO) twice for 2 h at 37 °C for no ferritin

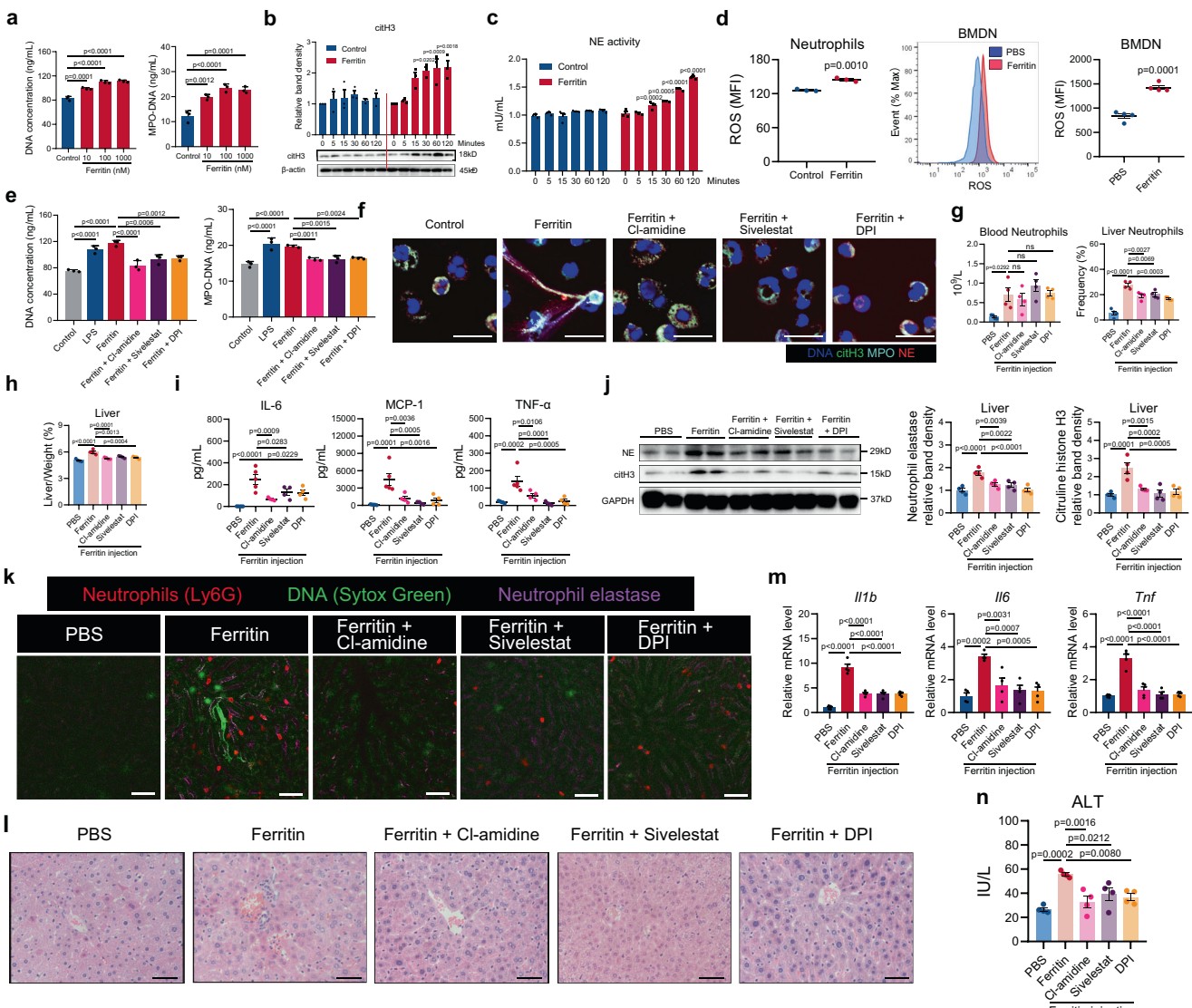

**Fig. 4 | Ferritin-induced NETosis is dependent on PAD4, NE and ROS.**
**a** Quantification of cell-free DNA and MPO-DNA complexes released by neutrophils from healthy donors stimulated with 10, 100 and 1000 nM ferritin. Plots show means ± SD from one representative experiment with $n = 3$ technical replicates. Results were confirmed in ten independent experiments for cell-free DNA and three independent experiments for MPO-DNA using cells from different donors. **b**, **c** Changes of citH3 levels (**b**) and NE activity (**c**) in healthy donors' neutrophils upon ferritin stimulation ($n = 3$ biologically independent experiments). Representative Western blotting of citH3 was shown. **d** Detection of ROS by flow cytometry in ferritin-stimulated human neutrophils (representative results from three independent experiments) and BMDNs from ferritin-treated mice ($n = 4$). **e**, **f** The inhibitory effects of PAD4, NE and ROS with corresponding inhibitors Cl-amidine, sivelestat and DPI on ferritin-induced NET formation were detected by cell-free DNA, MPO-DNA complexes (**e**) and immunofluorescence of MPO (cyan), NE (red), citH3 (green) and DNA (blue) (**f**) in neutrophils from healthy donors. Scale bars, 20 μm. Plots in (**e**) show means ± SD from one representative experiment with $n = 3$

technical replicates. Results in (**e**, **f**) were confirmed in three independent experiments using cells from different donors. **g**–**i** Peripheral blood and liver neutrophil (**g**, $n = 4$), changes of liver to body weight ratio (**h**, $n = 4$) and serum cytokines, IL-6, MCP-1 and TNF-α (**i**, $n = 5$ in PBS and ferritin groups, $n = 4$ in other groups) were evaluated after treatment with Cl-amidine, sivelestat or DPI in ferritin-treated mice at 6 h. **j** Western blotting analysis for NET markers citH3 and NE in the liver ($n = 4$). **k** Representative images of neutrophils infiltration (red) and NET release (DNA: green; NE: magenta) in the liver by intravital microscopy. One representative image of livers from one mouse per group was shown. Scale bars, 50 μm. **l**–**n** H&E staining of liver inflammatory infiltration (**l**), liver mRNA expression of *Il1b*, *Il6* and *Tnfa* (**m**) and serum ALT levels (**n**) in ferritin-injected mice at 6 h ($n = 4$). Scale bars, 50 μm. Data in (**b**–**d**, **g**–**j**, **m**, **n**) are presented as means ± SEM, Data in (**a** and **e**) are presented as means ± SD of 3 technical replicates. Ns, not significant. Unpaired two-sided Student's $t$ test (**d**), one-way ANOVA (**a**, **e**, **g**–**j**, **m**, **n**) and two-way ANOVA with Bonferroni's multiple comparison test (**b** and **c**). Source data are provided as a Source Data file.

---

blockers or neutralizing antibodies available in the current market. The sera before and after absorption were collected from 3 AOSD patients. 10% sera were added to stimulate the neutrophils from healthy control (HC) for 3.5 h at 37 °C.

### Isolation and stimulation of mouse bone marrow-derived neutrophils
BMDNs were isolated by density gradient centrifugation using Histopaque 1119 (11191, Sigma-Aldrich) and Histopaque 1077 (10771, Sigma-

Aldrich). Total bone marrow cells were collected from tibias and femurs and the RBCs were lysed. Mature neutrophils were purified by centrifugation for 30 min at $845 \times g$ without braking on a Histopaque 1119 and Histopaque 1077. The neutrophils were collected at the interface of the Histopaque 1119 and Histopaque 1077. Mice neutrophils ($1 \times 10^6$ cells/mL) cultured in RPMI 1640 supplemented with 10% FBS and were stimulated with ferritin (100 nM, F4503, Sigma-Aldrich), LPS (100 ng/mL, L4391, Sigma-Aldrich) or PMA (20 nM, P1585, Sigma-Aldrich) for 3.5 h at 37 °C.

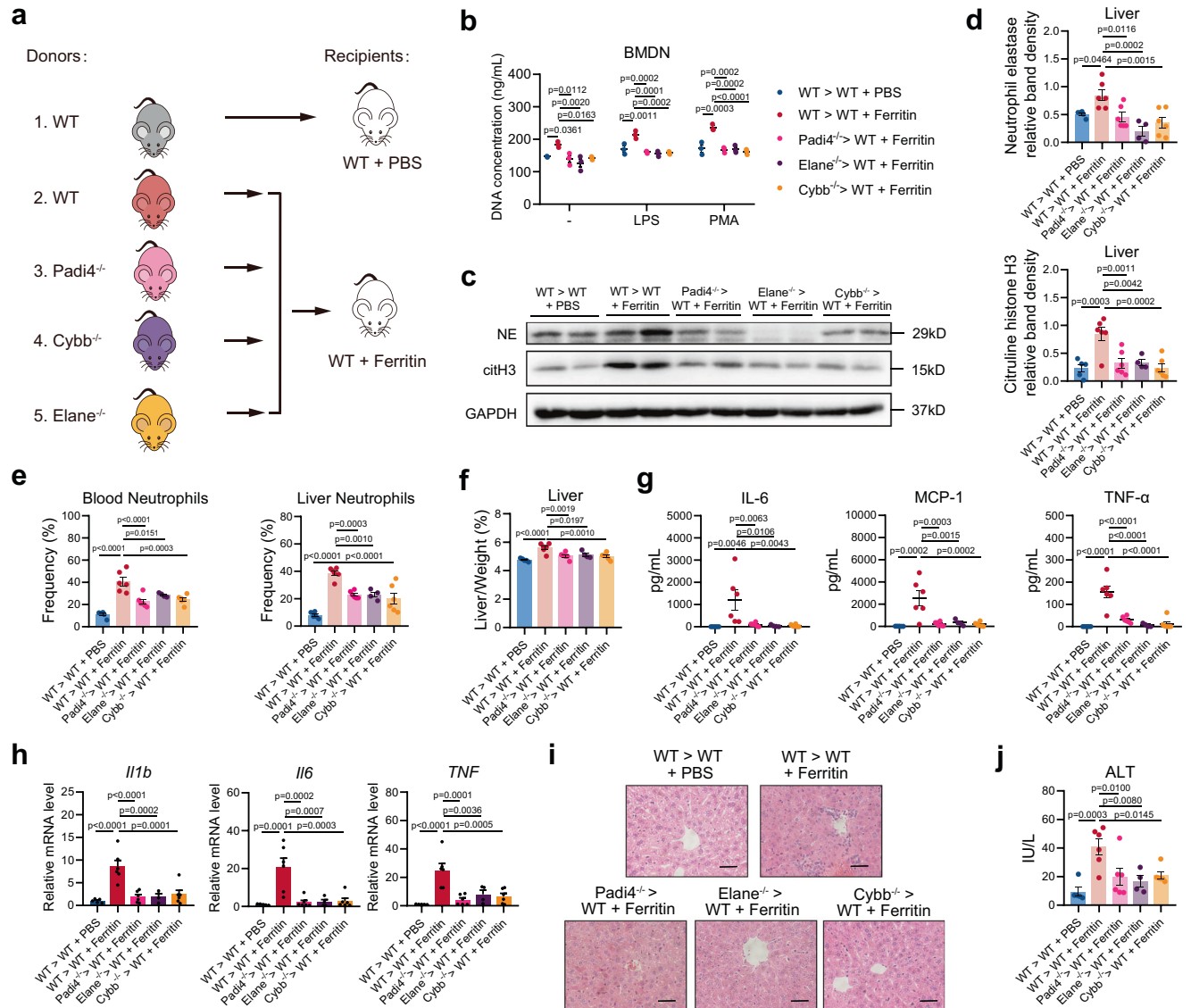

**Fig. 5 | Genetic disruption of PAD4, NE and ROS in BM improves ferritin-induced inflammation. a** Experimental overview: Bone marrow (BM) from WT, Padi4⁻/⁻, Elane⁻/⁻ and Cybb⁻/⁻ mice in C57BL6 background were transplanted to WT recipients and allowed to reconstitute for 4 weeks following which ferritin was injected. **b** Quantification of cell-free DNA released by BMDNs (*n* = 3). **c, d** Western blotting analysis for NET markers citH3 and NE in the livers (WT > WT + PBS: *n* = 5, WT > WT + Ferritin: *n* = 6, Padi4⁻/⁻ >WT + Ferritin: *n* = 6, Elane⁻/⁻ >WT + Ferritin: *n* = 4, Cybb⁻/⁻ >WT + Ferritin: *n* = 6). Representative Western blotting was shown. **e–g** Peripheral blood and liver neutrophil (**e**), changes of liver to body weight ratio (**f**)

and serum cytokines, IL-6, MCP-1 and TNF-α (**g**) were evaluated in mice transplanted with Padi4⁻/⁻, Elane⁻/⁻ and Cybb⁻/⁻BM (WT > WT + PBS: *n* = 5, WT > WT + Ferritin: *n* = 6, Padi4⁻/⁻ >WT + Ferritin: *n* = 6, Elane⁻/⁻ >WT + Ferritin: *n* = 4, Cybb⁻/⁻ >WT + Ferritin: *n* = 6). **h–j** liver mRNA expression of *Il1b*, *Il6* and *Tnfa* (**h**), H&E staining of liver inflammatory infiltration (**i**), and serum ALT levels (**j**) (WT > WT + PBS: *n* = 5, WT > WT + Ferritin: *n* = 6, Padi4⁻/⁻ >WT + Ferritin: *n* = 6, Elane⁻/⁻ >WT + Ferritin: *n* = 4, Cybb⁻/⁻ >WT + Ferritin: *n* = 6). Scale bars, 50 μm. Data are presented as means ± SEM; ns not significant. One-way ANOVA (**b, d, e–h, j**). Source data are provided as a Source Data file.

## Quantification of cell-free DNA and NETs-DNA complexes

Cell-free DNA was quantified using the Quant-iT PicoGreen double-stranded DNA (dsDNA) assay kit (P11496, Invitrogen) according to the manufacturer's instructions. 10% serum or cell culture supernatants was added per well, followed by incubation for 10 min. NE-DNA, MPO-DNA, and citH3-DNA complexes were quantified using the Quant-iT PicoGreen. As the capturing antibodies, anti-citH3 (1:1000, ab5103, Abcam), anti-NE (1:2000, ab68672, Abcam) or anti-MPO monoclonal antibodies (1:1000, ab25989, Abcam) were coated onto 96-well microtiter plates overnight at 4 °C. After blocking in 1% BSA for 90 min at room temperature, 10% serum or cell culture supernatants was added per well, followed by incubation overnight at 4 °C. PicoGreen was added to detect cell-free DNA and NET-DNA complexes.

## NET detection and quantification

After stimulation, neutrophils were fixed with 4% paraformaldehyde. Protein staining was performed using a rabbit polyclonal anti-citH3 antibody (1:200, ab5103, Abcam), a mouse monoclonal anti-NE antibody (1:50, sc-55549, Santa Cruz), and a goat monoclonal anti-MPO antibody (1:200, AF3667, R&D) overnight at 4 °C. After three washes, appropriate fluorochrome-conjugated secondary antibodies (1:200, Alexa Fluor 594-conjugated rabbit anti-mouse IgG, 33912ES60; 1:200, Alexa Fluor 488-conjugated goat Anti-rabbit IgG, 33106ES60; 1:200, Alexa Fluor 647-conjugated rabbit anti-goat IgG, 33713ES60; all from YEASEN, Shanghai, China) were applied for 1 h incubation at room temperature. DNA was stained with Hoechst 33342 (1:2000, H3570, Invitrogen) for 5 min. After three washes, images were obtained using FV3000 confocal system (Olympus).

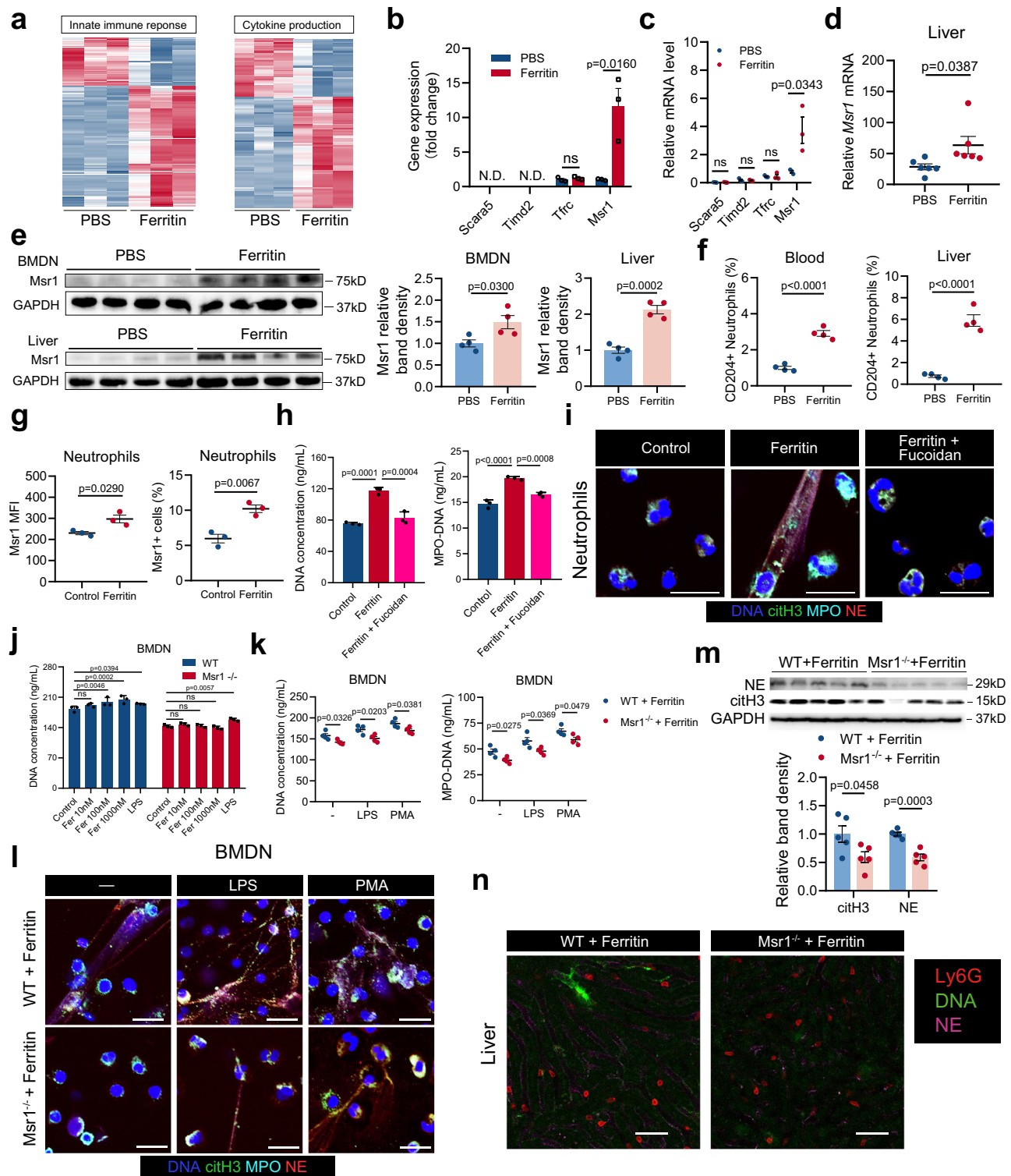

For quantification, total extracellular DNA generated by cultured neutrophils was digested with 30 mU/mL micrococcal nuclease (MNase, Thermo Fisher Scientific) for 10 min at 37 °C, and then stopped with 5 mM EDTA. Cell-free DNA and MPO-DNA complex in the supernatants was quantified by PicoGreen. For BMDNs, NET formation was confirmed by visualization using 0.3 μM SYTOX green (S7020, Invitrogen, USA) and images were captured using an Olympus microscope (IX73).

**Blood counts**
Mice were bled from the retro-orbital plexus after isoflurane anaesthesia. 50 μl blood was collected into an EDTA-containing tube. The

samples were analyzed using a SYSMEX Hematology Analyzer (pocH-100iVD, Sysmex, Japan).

**Serum cytokine and ALT measurement**
Serum was collected from whole blood by centrifugation at $845 \times g$ for 10 min. Serum ALT measurement was performed using a commercially available kit (700260, Cayman chemical company, MI). 150 μl of substrate, 20 μl of cofactor and 20 μl of positive control or sample were added to the wells. After incubation for 15 min at 37 °C, 20 μl of ALT initiator was added and the absorbance at 340 nm was measured for 10 min at 37 °C. The ALT

**Fig. 6 | Msr1 is responsible for ferritin-induced NET formation. a, b** Differentially expressed genes of "innate immune response" and "cytokine production" pathways (**a**), and gene expression of potential ferritin receptors in the BMDNs from PBS- and ferritin-treated mice (**b**) (*n* = 3). **c** mRNA levels of potential ferritin receptors were confirmed in the FACS-sorted BMDNs (*n* = 3). **d, e** Msr1 mRNA levels in the livers (**d**, *n* = 6) and protein levels in the BMDNs and livers (**e**, *n* = 4) were determined by qRT-PCR and Western blotting analysis. **f** Flow cytometry for Msr1⁺ neutrophils in the peripheral blood and liver in ferritin-treated mice (*n* = 4). **g** Msr1 mean fluorescence intensity (MFI) and positive cell percentage of human neutrophils after ferritin stimulation. Experiments were repeated three times. **h, i** The effects of Msr1 blockade on ferritin-induced NET formation were detected by cell-free DNA, MPO-DNA complexes (**h**) and immunofluorescence of MPO (cyan), NE (red), citH3 (green) and DNA (blue) (**i**) in neutrophils from healthy donors. Plots in (**h**) show means ± SD from one representative experiment with *n* = 3 technical replicates. Results in (**h** and **i**) were confirmed in three independent experiments using cells from different donors. Scale bars, 20 μm. **j** DNA release of BMDNs from WT and Msr1⁻/⁻ mice with ferritin stimulation. Plots show means ± SD from one representative experiment with *n* = 3 technical replicates. Results were confirmed in three independent experiments using cells from different mice. **k, l** Detection of cell-free DNA, MPO-DNA complexes (**k**) and immunofluorescence of MPO (cyan), NE (red), citH3 (green) and DNA (blue) (**l**) in the BMDNs from ferritin-treated WT and Msr1⁻/⁻ mice at 6 h (*n* = 4). Scale bars, 20 μm. **m** Western blotting analysis for NET markers citH3 and NE in the liver from ferritin-treated WT and Msr1⁻/⁻mice at 6 h (*n* = 5). **n** Representative images of neutrophils infiltration (red) and NET release (DNA: green; NE: magenta) in the liver of ferritin-treated WT and Msr1⁻/⁻ mice by intravital microscopy. One representative image of livers from two mice per group was shown. Scale bars, 50 μm. Data in (**b**–**g**, **k**, **m**) are presented as means ± SEM; Data in (**h** and **j**) are presented as means ± SD of 3 technical replicates. N.D. not detected, ns not significant. Unpaired two-sided Student's *t* test (**b**–**g**, **k** and **m**), one-way ANOVA (**h** and **j**). Source data are provided as a Source Data file.

value was determined by calculating the slope of the absorbance curve.

Serum IL-6, IL-10, TNF-α, IFN-γ, MCP-1 and IL-12p70 were measured with Cytokine Bead Array (552364, BD Biosciences, USA). 50 μl of the standard dilutions or samples were incubated with capture beads and detection reagent for 2 h at room temperature. Assays were performed by a FACS Canto II cytometer (BD).

### Histology, immunohistochemistry, and immunofluorescence of liver
Liver tissue sections were stained with H&E staining. To detect neutrophils, macrophages and lymphocytes in the liver tissue, paraffin-embedded mouse liver sections were stained by antibodies against Ly6G (1:1000, GB11229, Servicebio, Wuhan, China), F4/80 (1:1000, GB11027, Servicebio, Wuhan, China), CD3 (1:1000, GB11014, Servicebio, Wuhan, China) and B220 (1:4000, GB11066, Servicebio, Wuhan, China). To detect NET formation in the liver tissue, the sections were incubated with anti-citH3 (1:200, ab5103, Abcam), anti-NE (1:50, sc-55549, Santa Cruz) and anti-MPO (1:200, AF3667, R&D). 4′,6-Diamidino-2-phenylindole (2 μg/ml, DAPI, Servicebio, Wuhan, China) was used to detect DNA. Finally, slides were visualized using an Olympus microscope (IX73, Tokyo, Japan).

### RNA sequencing
Gradient-centrifuged BMDNs from PBS- and ferritin-treated mice were used for total RNA isolation. Oligo(dT)-attached magnetic beads-purified mRNA was fragmented into pieces at appropriate temperature. cDNA was generated by random hexamer-primed reverse transcription. Afterwards, RNA Index Adapters and A-Tailing Mix were added to end repair. The cDNA fragments were amplified by PCR, and products were purified using Ampure XP Beads. The double-strand PCR products were denatured and circularized by the splint oligo sequence to construct the final library. Then the single-strand circle DNA was formatted as the final library. The final library was amplified by phi29 to produce DNA nanoball (DNB) with more than 300 copies of one molecule. DNBs were loaded into patterned nanoarray and single end 50 bases reads were generated on BGIseq500 platform (BGI-Shenzhen, China). HISAT2 (v2.0.4) was used to map the clean reads to the genome. Bowtie2 (v2.2.5) was applied to align the clean reads to the reference coding gene set. RSEM (v1.2.12) was used to calculate the expression level of gene. Differentially expressed genes (DEGs) with a fold change >2 and *p* value < 0.05 were determined using DESeq2(v1.4.5).

### Quantitative real-time PCR
Total RNA was extracted using Trizol reagent following manufacturer's instructions (9109, Takara, Japan), and reverse-transcribed into cDNA using PrimeScript™ RT Reagent Kit (RR036, Takara). qRT-PCR was performed with SYBR Green (B21703, Bimake, China). The relative expression levels of mRNA were normalized against β-actin (mouse) or GAPDH (human). Specific primers of mouse Ly6G, IL-1β, IL-6, IL-10, TNF-α, EMR1, PPARγ, IFN-γ, ARG1, TGF-β, CD163, CD206, iNOS, Msr1, Scara5, Timd2, Tfrc and human Msr1 were used. Primer sequences were listed in Supplementary Table 2.

### Flow cytometry
Intrahepatic and blood leukocytes were stained for 15 min at 4 °C using fluorescently labeled antibodies: PerCP-conjugated anti-mouse CD45 (30-F11 clone, 1:100, 557235, BD), FITC-conjugated anti-mouse CD11b (M1/70 clone, 1:100, 101206, Biolegend), PE-Cy7-conjugated anti-mouse Ly6G (1A8 clone, 1:100, 560601, BD), APC-conjugated anti-mouse Ly6C (HK1.4 clone, 1:100, 128016, BD), FITC-conjugated anti-mouse CD3 (17A2 clone, 1:100, 561798, BD), PE-Cy7-conjugated anti-mouse CD4 (RM4-5 clone, 1:100, 552775, BD), APC-H7-conjugated anti-mouse CD8a (53-6.7 clone, 1:100, 560182, BD), APC-conjugated anti-mouse CD19 (1D3 clone, 1:100, 550992, BD), PE-conjugated anti-mouse CD49b (DX5 clone, 1:100, 553858, BD), PE-conjugated anti-mouse F4/80 (BM8 clone, 1:100, 123110, Biolegend) and Alexa Fluor 647-conjugated anti-mouse Msr1 (2F8 clone, 1:100, MCA1322A647, AbD Serotec, NC, USA). Human blood was stained with PE-conjugated anti-human CD11b (ICRF44 clone, 1:100, 555388, BD), PE-Cy7-conjugated anti-human CD66b (G10F5 clone, 1:100, 305116, Biolegend) and APC-conjugated anti-human Msr1 (7C9C20 clone, 1:100, 371905, Biolegend). All assays were performed by a FACS Canto II cytometer (BD). For cell sorting, FACSAria was used. Data were analyzed using FlowJo software (Tree Star, Inc., Ashland, OR).

### Detection of reactive oxygen species (ROS) production
DCFH-DA (1:1000, Beyotime Institute of Biotechnology, Shanghai, China) was used to detect intercellular ROS according to the manufacturer's instructions. The $5 \times 10^5$ cells in a final volume of 500 μl were incubated for 20 min with 10 μM DCFH-DA. The cells were washed with serum-free RPMI 1640 for 3 times. Flow cytometry was used for quantitative analysis. Data were analyzed using FlowJo software (Tree Star, Inc., Ashland, OR).

### Neutrophil elastase activity assay
NE activity was quantified using neutrophil elastase activity assay kit (600610, Cayman chemical). Standards or cell culture supernatants from human neutrophils treated with ferritin were plated in a 96-well plate. 10 μl of neutrophil elastase substrate was added to each well, and the plate was incubated at 37 °C for 1.5 h. Fluorescence was quantified at excitation and emission wavelengths of 485 nm and 525 nm, respectively.

### Intravital imaging of NET formation
Fluorescence imaging of NETs and neutrophils was performed with intravital imaging analysis. Mice were anaesthetized by an initial

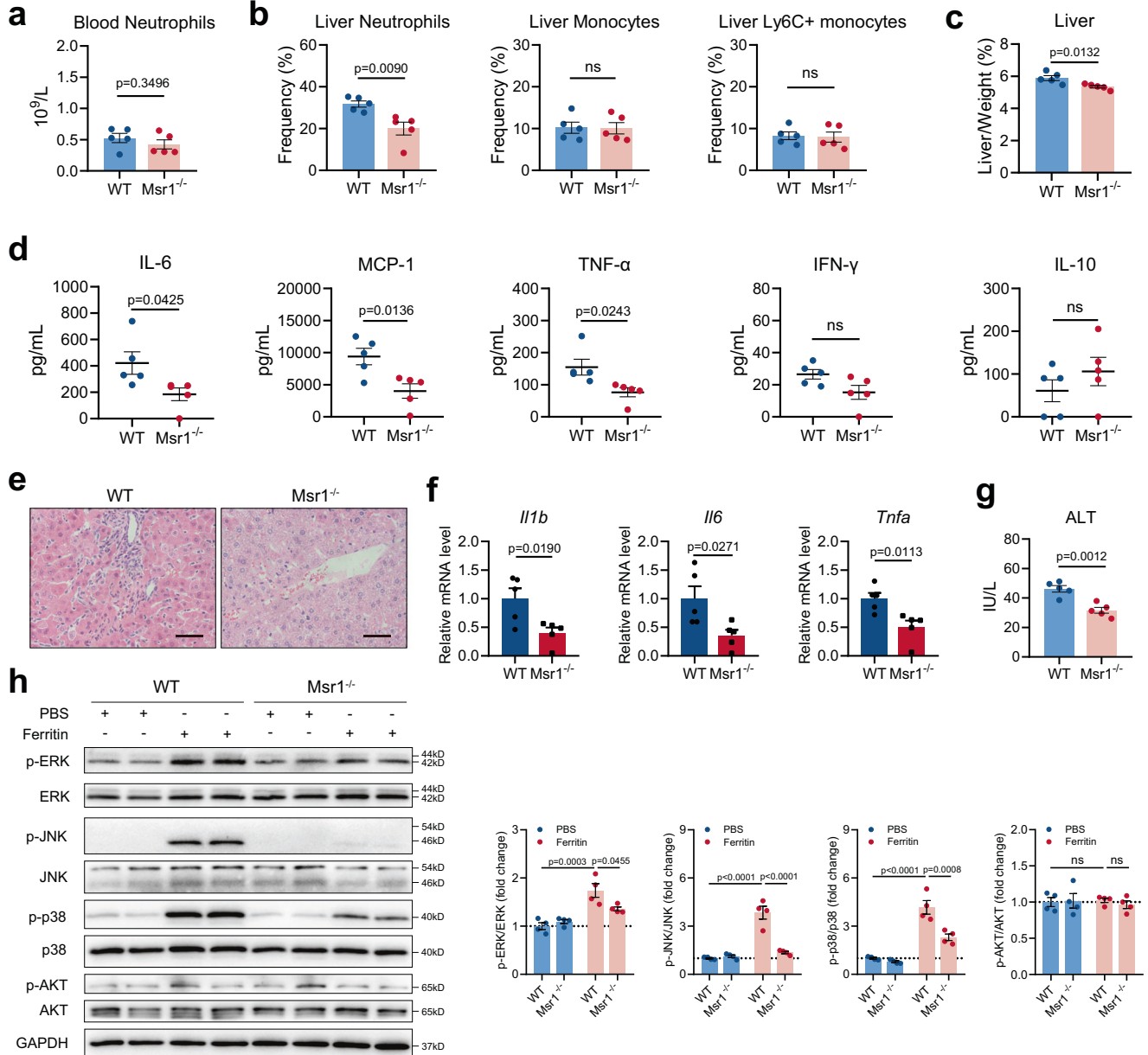

**Fig. 7 | Msr1 ablation ameliorates ferritin-induced inflammation. a–d** Peripheral neutrophil numbers (**a**), hepatic myeloid infiltration (**b**), liver to body weight ratio (**c**) and serum cytokines (**d**) were assessed in ferritin-treated WT and Msr1$^{-/-}$ mice at 6 h (*n* = 5). **e–g** Liver inflammatory infiltration (**e**), mRNA expression of *Il1b*, *Il6* and *Tnfa* (**f**) and serum ALT levels (**g**) were identified in WT and Msr1$^{-/-}$ mice at 6 h post ferritin injection (*n* = 5). Scale bars, 50 μm. **h** Western blotting analysis of ERK, JNK, p38, and Akt in the BMDNs from ferritin-treated WT and Msr1$^{-/-}$ mice at 6 h (*n* = 4). Densitometric analysis of p-ERK/total ERK, p-JNK/total JNK, p-p38/total p38 and p-Akt/total Akt was shown. Data are presented as means ± SEM; ns not significant. Unpaired two-sided Student's *t* test (**a–d**, **f**, and **g**) and two-way ANOVA with Bonferroni's multiple comparison test (**h**). Source data are provided as a Source Data file.

intraperitoneal injection of avertin (100 mg/kg). The tail vein was catheterized to allow delivery of fluorescent probes and to maintain anaesthesia as required. A heating pad was used to maintain the temperature of the mice at 37 °C. The exposed liver was bathed in normal saline and cover slipped. Neutrophils were visualized by injection of PE anti-mouse Ly6G (2.5 μg, 1A8 clone, 127608, Biolegend) 10 min before intravital imaging. Then NETs were visualized by co-staining of extracellular DNA (Sytox green, 5 μM, S7020, Invitrogen) and neutrophil elastase (Alexa Fluor 647 anti-NE antibodies, 1.6 μg, sc-55549 AF647, Santa Cruz). Images and videos were acquired using Olympus FV3000 inverted microscope with a Galvano scanner and 20×/NA 0.75 UPLANSAPO objective lenses. Sytox green, PE, and Alexa Fluor 647 were irradiated using 488, 561 and 640 nm laser lines, respectively. 20 min videos were recorded every 20 s after ferritin injection for 6 h.

Fiji software on Image J (v1.53c, Bio-Rad, USA) was used to create movies in 6 h control group and 6 h ferritin group.

### Detection of NET formation in human liver biopsy
Liver biopsy from patients with AOSD (*n* = 1) was analyzed for NET formation by immunofluorescence using anti-citH3 (1:200, ab5103, Abcam) and anti-MPO (1:1000, ab25989, Abcam). The liver biopsy of healthy control (*n* = 1) is obtained from a patient with haemangioma during surgery with normal liver histology, which could be used as the normal liver tissue according to previous studies[50].

### Immunoblotting
Human neutrophils, mouse livers, and BMDNs were lysed in RIPA lysis buffer (Beyotime Institute of Biotechnology, Shanghai, China)

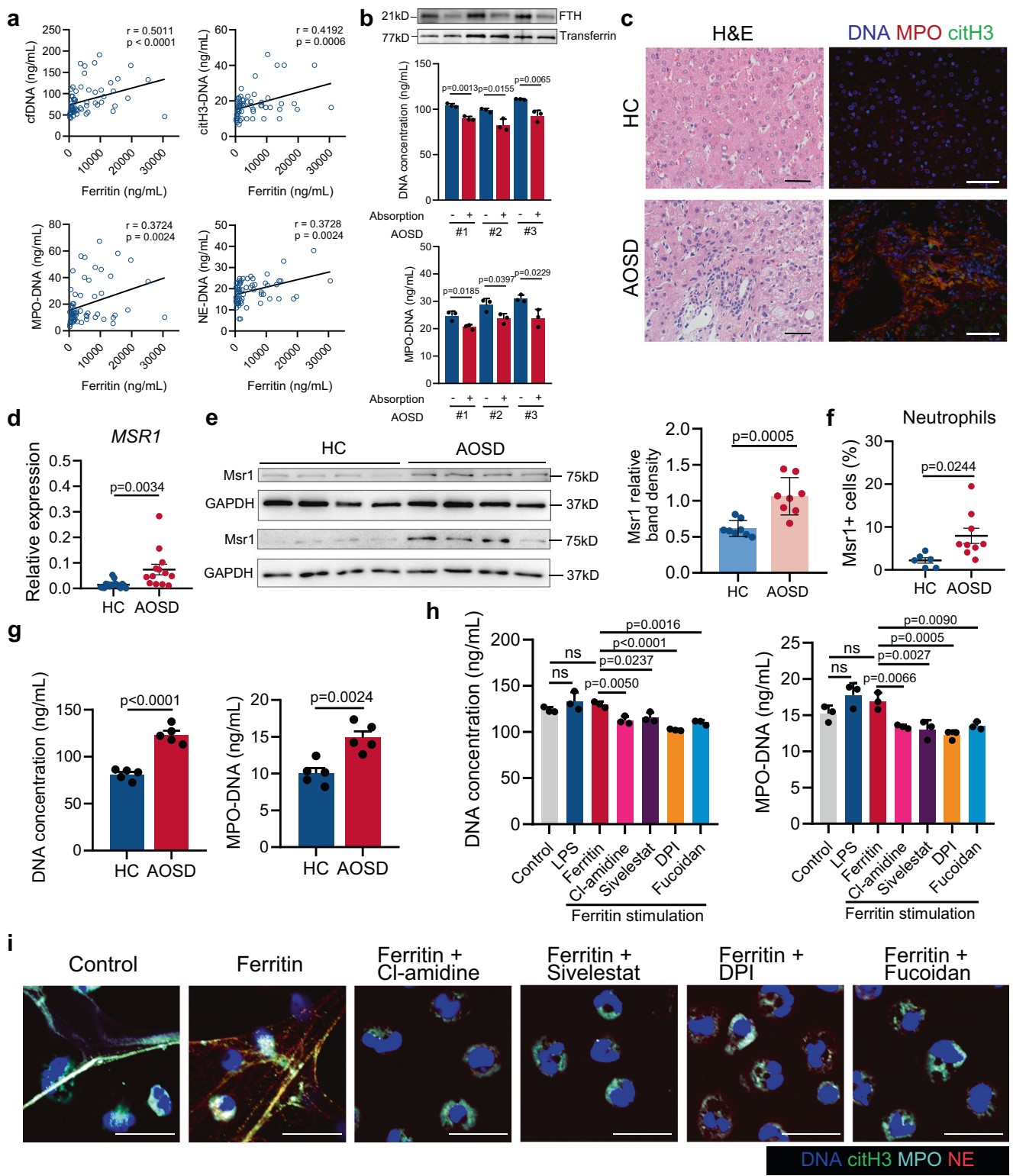

containing protease inhibitor cocktails (Roche Diagnostics, Mannheim, Germany) and phosSTOP (Roche Diagnostics, Mannheim, Germany). The protein lysates (20–40 µg protein) were separated on 10% SDS/PAGE gel and transferred onto polyvinylidene fluoride membrane. Membranes were incubated with primary antibodies against rabbit anti-ERK (1:1000, 137F5 clone, 4695, CST), rabbit anti-phospho-ERK (1:1000, D13.14.4E clone, 4370, CST), rabbit anti-JNK (1:1000, 56G8 clone, 9258, CST), rabbit anti-phospho-JNK (1:1000, 81E11 clone, 4668, CST), rabbit anti-p38 (1:1000, D13E1 clone, 8690,

CST), rabbit anti-phospho-p38 (1:1000, D3F9 clone, 4511, CST), rabbit anti-AKT (1:1000, C67E7 clone, 4691, CST), rabbit anti-phospho-AKT (1:1000, C31E5E clone, 2965, CST), rabbit anti-Msr1 (1:1000, ab151707, Abcam), rabbit anti-NE (1:1000, ab68672, Abcam) and rabbit anti-citH3 (1:1000, ab5103, Abcam) overnight at 4 °C followed by HRP-conjugated anti-rabbit IgG (1:5000, 7074 S, CST) or HRP-conjugated anti-mouse IgG (1:5000, L3032, Signalway Antibody, Maryland, USA), and the signals were detected by ECL assays (WBKLS0500, Millipore, USA). Anti-GAPDH (1:1000, AF1186, Beyotime Institute of

**Fig. 8 | Hyperferritinemia contributes to increased NET formation in AOSD patients. a** The correlations between serum ferritin levels and circulating NET-DNA complexes and cell-free DNA levels in AOSD patients ($n = 64$). **b** Ferritin (FTH) in sera from 3 patients with active AOSD was either absorbed away (after absorption) or not absorbed (before absorption), and then measured by Western blotting. Quantification of cell-free DNA and MPO-DNA complexes released by neutrophils from healthy donors stimulated with sera above. Plots show means ± SD from one representative experiment with $n = 3$ technical replicates. Results were confirmed in three independent experiments using cells from different donors. **c** H&E staining for inflammatory infiltration and immunofluorescence detection of NET markers, including citH3, MPO and DNA, in liver biopsies from healthy donor ($n = 1$) and AOSD patient ($n = 1$). Scale bars, 50 μm. **d–f** Msr1 expression on neutrophils from healthy donors and AOSD patients measured by qRT-PCR (**d**), Western blotting (**e**) and flow cytometry (**f**). ($n = 17$ in HC and 13 in AOSD for qRT-PCR; $n = 8$ per group

for Western blotting, $n = 6$ in HC and 9 in AOSD for flow cytometry) (**g**) The basal NET formation levels were detected by cell-free DNA and MPO-DNA complexes in neutrophils from healthy controls ($n = 5$) and AOSD patients ($n = 5$). **h, i** The inhibitory effects of PAD4, NE, ROS and Msr1 inhibitors on ferritin-induced NET formation were detected by cell-free DNA, MPO-DNA complexes (**h**) and immunofluorescence of MPO (cyan), NE (red), citH3 (green) and DNA (blue) (**i**) in neutrophils from patients with AOSD. Plots in h show means ± SD from one representative experiment with $n = 3$ technical replicates. Results in (**h** and **i**) were confirmed in three independent experiments using cells from different donors. Scale bars, 20 μm. Data in (**d–g**) are presented as means ± SEM; Data in (**b** and **h**) are presented as means ± SD of 3 technical replicates. ns not significant. HC healthy control. Spearman's nonparametric test (**a**), unpaired two-sided Student's $t$ test (**b**, **d–f**, and **g**) and one-way ANOVA with Bonferroni's multiple comparison test (**h**). Source data are provided as a Source Data file.

---

Biotechnology, Shanghai, China) or β-actin antibody (1:1000, 3700 S, CST) was used as an internal control. Bands were quantitated using Image J (v1.48 & v1.53c, Bio-Rad, USA), and results are expressed as fold change relative to the internal control.

### Statistical analysis
All data were statistically analyzed using the SPSS version 20.0 (SPSS Inc., Chicago, IL, USA) and Graphpad Prism v8.0 software. Quantitative data are expressed as the means ± SEM (standard error of the mean) or means ± SD (standard deviation) as indicated. Data with a Gaussian distribution was analyzed using an unpaired two-sided $t$-test, one-way or two-way analysis of variance (ANOVA), while nonparametric data were assessed using the Mann–Whitney U test or Wilcoxon rank-sum test. Bonferroni post hoc tests were used to compare all pairs of treatment groups when the overall $p$ value was <0.05. All tests were two-sided, and $p$ values < 0.05 were considered statistically significant.

### Reporting summary
Further information on research design is available in the Nature Portfolio Reporting Summary linked to this article.

## Data availability
The RNA sequencing data generated in this study have been deposited in the GenBank (Gene Expression Omnibus; GEO) under accession code GSE179679. All the data supporting the findings of this study are available within the article and its supplementary information files, or can be obtained from the corresponding author upon reasonable request. Source data are provided with this paper.

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

## Acknowledgements

This work was supported by the National Natural Science Foundation of China (82001704 to Q.H., 82171769 to C.Y., 31872737 to J.W.), Shanghai Sailing Program (20YF1427100 to Q.H., 22YF1425600 to J.J.), Shanghai Pujiang Young Rheumatologists Training program (SPROG201901 to Q.H., SPROG2109 to J.J.), and Shanghai Science and Technology Innovation Action (20Y11911500 to C.Y.).

## Author contributions

J.J., M.W., J.M., Y.M. performed main experiments, corresponding data analysis and wrote the paper. Y.W., N.M., J.T., D.Z., H.S., Y. Sun., H.L., and X. Cheng. elaborated the clinical database and analyzed data. Y. Su., J.Y., H.C., T.L., Z.Z., L.W., X. Chen., F.W., and H.Z. discussed the experimental strategy and performed statistical analyses. J.B. and J.W. corrected the paper. C.Y. and Q.H. designed the study and revised the paper. All authors approved the final paper.

## Competing interests

The authors declare no competing interests.

## Additional information

Jinchao Jia[1,4], Mengyan Wang[1,4], Jianfen Meng[1,4], Yuning Ma[1,4], Yang Wang[2], Naijun Miao[2], Jialin Teng[1], Dehao Zhu[1],
Hui Shi [1], Yue Sun [1], Honglei Liu[1], Xiaobing Cheng[1], Yutong Su[1], Junna Ye[1], Huihui Chi [1], Tingting Liu [1],
Zhuochao Zhou[1], Liyan Wan[1], Xia Chen[1], Fan Wang[1], Hao Zhang[1], Jingjing Ben [3] ✉, Jing Wang [2] ✉,
Chengde Yang [1] ✉ & Qiongyi Hu [1] ✉

[1]Department of Rheumatology and Immunology, Ruijin Hospital, Shanghai Jiao Tong University School of Medicine, Shanghai, China. [2]Shanghai Institute of
Immunology, Department of Immunology and Microbiology, Shanghai Jiao Tong University School of Medicine, Shanghai, China. [3]Department of Patho-
physiology, Key Laboratory of Cardiovascular Disease and Molecular Intervention, Nanjing Medical University, Nanjing, China. [4]These authors contributed
equally: Jinchao Jia, Mengyan Wang, Jianfen Meng, Yuning Ma. ✉e-mail: bjj@njmu.edu.cn; jingwang@shsmu.edu.cn; yangchengde@sina.com; huqion-
gyi131@163.com

