## [Peer Review File · Nature Communications]

Ferritin triggers neutrophil extracellular trap-mediated cytokine storm through Msr1 contributing to adult-onset Still's disease pathogenesisREVIEWER COMMENTS

Reviewer #1 (Remarks to the Author):

In this detailed manuscript, Jia et al. test the role of neutrophils and neutrophil extracellular traps in ferritin-induced inflammation as a model of autoinflammatory diseases. They show that ferritin activates neutrophils in vitro and triggers an in vivo inflammatory disease which is attenuated by neutrophil depletion. Ferritin-induced responses are attenuated by inhibitors of NET formation and by deletion of the putative ferritin receptor, Msr1. The authors also link ferritin, NETs and Msr1 to human adult-onset Still's disease.

The experiments are well designed and carefully executed. The conclusions are mostly novel and mostly supported by the data. However, a few additional mechanistic studies and more direct link to COVID-19 should be provided.

Specific points:

- 1) The link to NET formation is a bit vague since the inhibitors used are of limited specificity. The authors should test knockouts of PAD4, NE and NOX2 in their model. Since all those molecules are tentatively expressed by neutrophils, testing bone marrow chimeras with knockout hematopoietic cell should suffice, simplifying the logistics of the experiments.
- 2) The authors mention COVID-19 as a putative hyperferritinaemic condition. They should more directly link their model to COVID-19 by testing sera/neutrophils from COVID-19 patients.
- 3) Since gradient-separated mouse neutrophil preparations usually contain a large number of contaminating cells with much higher transcriptional activity, gene expression changes should be confirmed using FACS-sorted neutrophils.
- 4) The effect of ferritin is often rather modest. This should be discussed in more detail.

Reviewer #2 (Remarks to the Author):

well done and well written. Finally an explanation for the presence of high ferritin and its insruamntal roel in th inflammatory reaction.Th epaper can be accepted as si and may deserve an editorial,

Reviewer #3 (Remarks to the Author):

This is a very nice investigation elucidating a Msr1 dependent neutrophil NETosis mechanism explaining the liver injury and hypercytokinemia seen in hyperferritinemic syndromes which include Adult Onset Still's Disease , Macrophage Activation Syndrome, Hyperferritinemic Sepsis/Septic Shock, Anti-Phospholipid Syndrome, and some cases of COVID19.

The investigators further provide evidence for this mechanism in a small number of AOSD patients

I think this is a very important finding with potentially important clinical implications especially since Msr1 inhibitors including the Brown Seaweed component fucoidan (ahelat5h food supplement) are not presently being used to treat these patients and could be easily studied and potentially safely given world wide and rather inexpensively so.

My suggestions

- 1) Soften the title to 'Ferritin triggers neutrophil extracellular traps-mediated cytokine storm through Msr1, importance in Adult-Onset Still's Disease'
- 2) To assure reproducibility by future investigators of this work please list the materials used more specifically, especially the ferritin form used (Sigma Aldrich but which ferritin?), the ferritin removal column used?, the availble source for Msr-/- mice.
- 3) Change 'we revel' to 'we reveal' page 14, line 5

Very Nice Work

Summary of Changes:

We thank the reviewers for their time and careful analysis of our manuscript. Our revised manuscript includes extensive data to address reviewer comments. Among these data, we have included a new Figure [Fig.5] to validate the effects of PAD4, NE and NOX2 in ferritin-induced inflammation. We performed bone marrow transplants of WT, Padi4^{-/-}, Elane^{-/-} and Cybb^{-/-} mice and confirmed our model in these BMT mice. Besides, we confirmed the gene expression changes of potential ferritin receptors using FACS-sorted neutrophils in new Fig. 6c.

To assure reproducibility, we have extended the Method section, including the sources of reagents, the dilution concentration of antibodies, and the detailed procedures of all experiments. We also provided a file named 'source data' containing the raw data of each figure. In addition, in the previous experiment, qRT-PCR was performed in batches. In order to improve the consistency, we pooled the mouse livers in the same time to run qRT-PCR again [new Fig. 3h and Fig. 4m].

We believe these new data prompted by reviewers improve the quality of our revised manuscript.

Responses to individual referees:

Reviewer #1 (Remarks to the Author):

In this detailed manuscript, Jia et al. test the role of neutrophils and neutrophil extracellular traps in ferritin-induced inflammation as a model of autoinflammatory diseases. They show that ferritin activates neutrophils in vitro and triggers an in vivo inflammatory disease which is attenuated by neutrophil depletion. Ferritin-induced responses are attenuated by inhibitors of NET formation and by deletion of the putative ferritin receptor, Msr1. The authors also link ferritin, NETs and Msr1 to human adult-onset Still's disease.

The experiments are well designed and carefully executed. The conclusions are mostly novel and mostly supported by the data. However, a few additional mechanistic studies and more direct link to COVID-19 should be provided.

Specific points:

1) The link to NET formation is a bit vague since the inhibitors used are of limited specificity. The authors should test knockouts of PAD4, NE and NOX2 in their model. Since all those molecules are tentatively expressed by neutrophils, testing bone marrow chimeras with knockout hematopoietic cell should suffice, simplifying the logistics of the experiments.

We greatly appreciated your constructive comments. We agree that testing knockouts of PAD4, NE and NOX2 is more specific and reliable than using inhibitors. Therefore, we performed bone marrow transplants of WT, Padi4^{-/-},

Elane^{-/-} and Cybb^{-/-} mice (new Fig. 5a). We found BMDNs from mice transplanted with Padi4^{-/-}, Elane^{-/-} or Cybb^{-/-} BM displayed lower levels of NET release after ferritin injection (new Fig. 5b). We also observed less NET deposition in the livers of mice transplanted with Padi4^{-/-}, Elane^{-/-} or Cybb^{-/-} BM after ferritin injection (new Fig. 5c-d). Deletion of these genes also resulted in decreased peripheral neutrophil frequency and hepatic neutrophil infiltration (new Fig. 5e), reduced hepatomegaly (new Fig. 5f), and suppression of serum cytokine levels and liver inflammation (new Fig. 5g-j). These results suggested ferritin-induced NET formation and systemic inflammation is PAD4, NE and NOX2- dependent.

2) The authors mention COVID-19 as a putative hyperferritinaemic condition. They should more directly link their model to COVID-19 by testing sera/neutrophils from COVID-19 patients.

We would like to express our gratitude towards the reviewer for an excellent suggestion which helped us to elevate the quality of our work substantially. As you stated, testing sera/neutrophils from COVID-19 patients would provide more convincing evidence. But regretfully, our lab conditions could not meet the requirements of COVID-19 study, we could not detect the relationship between ferritin levels and neutrophils in patients with COVID-19. Although we have not tested the importance of ferritin for NET formation in COVID-19, the ferritin-NETs-cytokine storm loop may be validated in the future as you suggested. As enhanced NET formation and hyperferritinemic state are features of patients with severe COVID-19 (Barnes, B.J. et al. *J Exp Med* 2020; Meng JF. et al. *Frontiers in Immunology* 2021). Also, we added this opinion in the Discussion section page 14, line 22 and page 15, line 1-3.

References:

1. Barnes, B. J. et al. Targeting potential drivers of COVID-19: Neutrophil extracellular traps. *J Exp Med* 217, doi:10.1084/jem.20200652 (2020).
2. Meng, J. et al. Cytokine Storm in Coronavirus Disease 2019 and Adult-Onset Still's Disease: Similarities and Differences. *Front Immunol* 11, 603389, doi: 10.3389/fimmu.2020.603389 (2020).

3) Since gradient-separated mouse neutrophil preparations usually contain a large number of contaminating cells with much higher transcriptional activity, gene expression changes should be confirmed using FACS-sorted neutrophils.

Thank you for making this important point. To address this, we confirmed the gene expression changes of potential ferritin receptors in FACS-sorted

neutrophils. As shown in new Fig. 6c, qRT-PCR showed a similar trend. Msr1 expression was significantly increased in BMDNs after ferritin treatment.

4) The effect of ferritin is often rather modest. This should be discussed in more detail.

We agree with the reviewer and have discussed in more detail in our revised manuscript. Ferritin is an iron storage protein and was previously considered as an acute-phase protein only as a biomarker of systemic inflammation. The aim of our study is to explore the role of ferritin in inflammation, especially the interaction between ferritin and neutrophils.

Reviewer #2 (Remarks to the Author):

well done and well written. Finally an explanation for the presence of high ferritin and its instrumental role in the inflammatory reaction. The paper can be accepted as is and may deserve an editorial,

Thank you for your inspiring and encouraging comments. In the revision, we have added bone marrow transplant experiments to validate our previous conclusion. We hope these new changes will improve our manuscript.

Reviewer #3 (Remarks to the Author):

This is a very nice investigation elucidating a Msr1 dependent neutrophil NETosis mechanism explaining the liver injury and hypercytokinemia seen in hyperferritinemic syndromes which include Adult Onset Still's Disease , Macrophage Activation Syndrome, Hyperferritinemic Sepsis/Septic Shock, Anti-Phospholipid Syndrome, and some cases of COVID19.

The investigators further provide evidence for this mechanism in a small number of AOSD patients

I think this is a very important finding with potentially important clinical implications especially since Msr1 inhibitors including the Brown Seaweed component fucoidan (a health food supplement) are not presently being used to treat these patients and could be easily studied and potentially safely given world wide and rather inexpensively so.

My suggestions

1) Soften the title to 'Ferritin triggers neutrophil extracellular traps-mediated cytokine storm through Msr1, importance in Adult-Onset Still's Disease'

Thank you very much for your constructive recommendation, and we truly agree with it. The submission instructions require titles should not contain punctuation or puns and should be 15 words or fewer, so we did not change it as the same. As suggested, we soften it to 'Ferritin triggers neutrophil extracellular traps-mediated cytokine storm through Msr1 with importance in Adult-Onset Still's Disease'.

2) To assure reproducibility by future investigators of this work please list the materials used more specifically, especially the ferritin form used (Sigma Aldrich but which ferritin?), the ferritin removal column used?, the available source for Msr^{-/-} mice.

We deeply appreciate the valuable comment. We carefully revised the method section to improve the reproducibility of our investigation. All antibodies and reagents used are now provided in the Methods section. Gating strategies of flow cytometry are now presented in new Supplementary Figure 1,2 and 7. Msr1-deficient (Msr1^{-/-}) mice on a C57BL/6 background were provided from Prof. Jingjing Ben in Nanjing Medical University (Nanjing, China), which were purchased from Jackson Laboratory. We thank the reviewer for this suggestion, as these changes have improved the quality of our research.

3) Change 'we revel' to 'we reveal' page 14, line 5

We greatly thank you for noting this. We have corrected it as recommended and the revised manuscript has been thoroughly checked for grammatical errors.

REVIEWERS' COMMENTS

Reviewer #1 (Remarks to the Author):

The authors have appropriately addressed my concerns.

Reviewer #3 (Remarks to the Author):

This is review of the revisions

The authors have answered most of my concerns

Please state in the methods section that that ferritin (F4503, Sigma -Aldrich, USA) is filtered and is form equine spleen which is a combination of H-Ferritin and L-Ferritin

Please mention in your limitations section that you have not tested other Sigma-Aldrich ferritins such as human ferritin, nor mouse ferritin

Responses to individual referees:

Reviewer #1 (Remarks to the Author):

The authors have appropriately addressed my concerns.

Thank you for taking the time to review our manuscript and all your insightful comments.

Reviewer #3 (Remarks to the Author):

This is review of the revisions

The authors have answered most of my concerns

Please state in the methods section that that ferritin (F4503, Sigma -Aldrich, USA) is filtered and is form equine spleen which is a combination of H-Ferritin and L-Ferritin

Please mention in your limitations section that you have not tested other Sigma-Aldrich ferritins such as human ferritin, nor mouse ferritin

Thank you for all your comments and your detailed review. We have added a statement in the Methods section that ferritin is sterile-filtered, derived from equine spleen, and composed of heavy chains and light chains.

In addition, following your suggestion, we have added a paragraph in the Discussion section to state our limitations.